# Mechanical inhibition of isolated $V_o$ from V/A-ATPase for proton conductance

Jun-ichi Kishikawa[1,2†], Atsuko Nakanishi[1,3†], Aya Furuta[1], Takayuki Kato[2,4], Keiichi Namba[4,5,6], Masatada Tamakoshi[7], Kaoru Mitsuoka[3], Ken Yokoyama[1]*

[1]Department of Molecular Biosciences, Kyoto Sangyo University, Kamigamo-Motoyama, Kyoto, Japan; [2]Institute for Protein Research, Osaka University, Suita, Japan; [3]Research Center for Ultra-High Voltage Electron Microscopy, Osaka University, Research Center for Ultra-High Voltage Electron Microscopy, Mihogaoka, Osaka, Japan; [4]Graduate School of Frontier Biosciences, Osaka University, Suita, Japan; [5]RIKEN Center for Biosystems Dynamics Research and SPring-8 Center, Suita, Japan; [6]JEOL YOKOGUSHI Research Alliance Laboratories, Osaka University, Suita, Japan; [7]Department of Molecular Biology, Tokyo University of Pharmacy and Life Sciences, Horinouchi, Hachioji, Tokyo, Japan

**Abstract** V-ATPase is an energy converting enzyme, coupling ATP hydrolysis/synthesis in the hydrophilic $V_1$ domain, with proton flow through the $V_o$ membrane domain, via rotation of the central rotor complex relative to the surrounding stator apparatus. Upon dissociation from the $V_1$ domain, the $V_o$ domain of the eukaryotic V-ATPase can adopt a physiologically relevant auto-inhibited form in which proton conductance through the $V_o$ domain is prevented, however the molecular mechanism of this inhibition is not fully understood. Using cryo-electron microscopy, we determined the structure of both the *holo* V/A-ATPase and isolated $V_o$ at near-atomic resolution, respectively. These structures clarify how the isolated $V_o$ domain adopts the auto-inhibited form and how the *holo* complex prevents formation of the inhibited $V_o$ form.

*For correspondence:
yokoken@cc.kyoto-su.ac.jp

†These authors contributed equally to this work

**Competing interests:** The authors declare that no competing interests exist.

## Introduction

Rotary ATPase/ATP synthases, roughly classified into F type and V type ATPases, are marvelous, tiny rotary machines (*Yokoyama and Imamura, 2005*; *Kinosita, 2012*; *Forgac, 2007*; *Yoshida et al., 2001*; *Kühlbrandt, 2019*). These rotary motor proteins share a basic molecular architecture composed of a central rotor complex and surrounding stator apparatus. These proteins function to couple ATP hydrolysis/synthesis in the hydrophilic $F_1/V_1$ moiety with proton translocation through the membrane embedded hydrophobic $F_o/V_o$ moiety by rotation of the central rotor complex relative to the surrounding stator apparatus, via a rotary catalytic mechanism (*Figure 1*; *Kinosita, 2012*; *Forgac, 2007*; *Yoshida et al., 2001*; *Kühlbrandt, 2019*; *Guo and Rubinstein, 2018*).

Thus, both F and V type ATPases are basically capable of either ATP synthesis coupled to the proton motive force (*pmf*) driven by the membrane potential or proton pumping powered by ATP hydrolysis. The F type ATPase (F-ATPase, or $F_oF_1$) in mitochondria functions as an ATP synthase coupled to respiration, whilst in some bacteria the complex can function as an ATP dependent proton pump (*Shibata et al., 1992*; *Kullen and Klaenhammer, 1999*).

The V type ATPase (V-ATPase, or $V_oV_1$) resides mainly in the membranes of acidic vesicles in eukaryote cells, functioning as a proton pump using a rotary catalytic mechanism (*Forgac, 2007*; *Yokoyama et al., 2003*; *Imamura et al., 2003*). Eukaryotic V-ATPases probably evolved from the prokaryotic enzymes (*Gogarten et al., 1989*; *Tsutsumi et al., 1991*), termed Archaeal ATPases or V/A-ATPases (*Forgac, 2007*; *Kühlbrandt and Davies, 2016*). The V/A-ATPase from a thermophilic bacterium, *Thermus thermophilus* (*Tth* V/A-ATPase) is a rotary ATPase that has been well

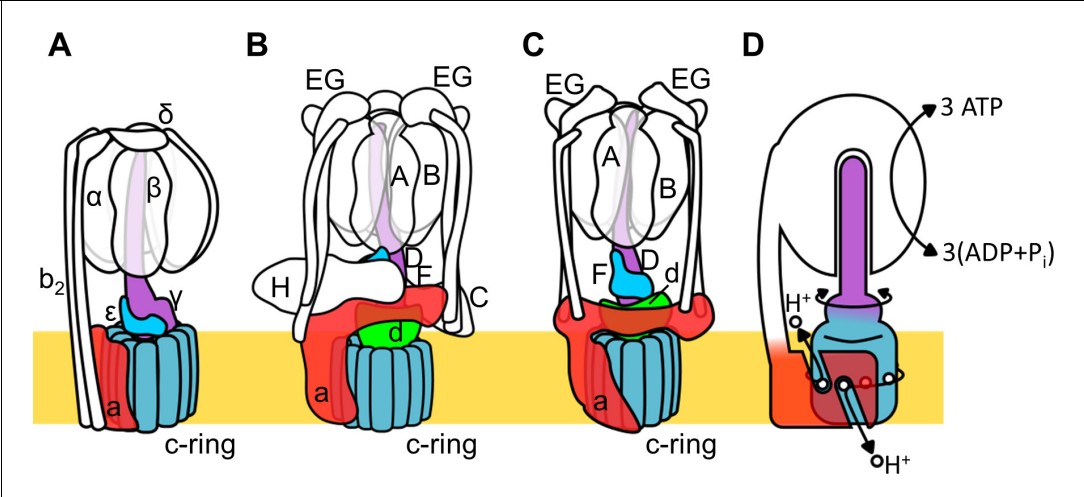

**Figure 1.** Schematic of rotary ATPase/synthases and the rotary catalytic mechanism. (**A**) Bacterial $F_oF_1$, (**B**) yeast V-ATPase, (**C**) *Tth* V/A-ATPase, (**D**) a schematic model of the rotary catalytic mechanism. The subunits of the central rotor complex are colored as follows: c-ring, dark blue; a-subunit, red; central axis, purple and cyan; and d-subunit, green.

The online version of this article includes the following figure supplement(s) for figure 1:

**Figure supplement 1.** Schematic representation of reversible dissociation of the $V_1$ domain induced by glucose depletion in yeast (**A**), and the assembly pathway of the *holo* V/A-ATPase in *Tth* cells (**B**).

characterized using both structure and single molecular observation studies (*Yokoyama and Imamura, 2005*; *Yokoyama et al., 2003*; *Imamura et al., 2003*; *Iwata et al., 2004*; *Makyio et al., 2005*; *Toei et al., 2007*; *Nakanishi et al., 2018*; *Schep et al., 2016*). The overall structure of *Tth* V/A-ATPase closely resembles that of the eukaryotic V-ATPase although it lacks some of the accessary subunits of the eukaryotic enzyme (*Figure 1B,C*). The *Tth* $V_1$ moiety is composed of four subunits with a stoichiometry of $A_3B_3D_1F_1$ and it is responsible for ATP synthesis or hydrolysis (*Yokoyama et al., 1998*; *Yokoyama et al., 1990*). Upon dissociation from $V_o$, the isolated $V_1$ moiety displays only ATP hydrolysis activity accompanied by rotation of the DF shaft. The *Tth* $V_o$ moiety, responsible for proton translocation across the membrane, contains a central rotor complex ($d_1c_{12}$) and stator apparatus made up of the *a* subunit and two EG peripheral stalks ($a_1E_2G_2$). In the *holo Tth* V/A-ATPase, *pmf* drives rotation of the $d_1c_{12}$ rotor complex relative to the surrounding stator, resulting in rotation of the entire central rotor complex ($D_1F_1d_1c_{12}$) and inducing sequential conformation changes in the $A_3B_3$ catalytic hexamer to produce three ATP molecules from ADP and inorganic phosphates per one rotation (*Figure 1D*).

The eukaryotic V-ATPase is regulated by a unique mechanism involving dissociation/association of $V_1$, that is likely to be a key factor in controlling the pH of acidic vesicles (*Kane and Parra, 2000*; *Sharma et al., 2019*; *Toei et al., 2010*). In yeast, glucose depletion in the culture medium induces dissociation of the $V_1$ domain from $V_o$ domain, resulting in reduced proton pumping activity of the V-ATPase (*Figure 1—figure supplement 1A*). It is likely that dissociated $V_o$ loses the ability to translocate protons as a result of auto-inhibition (*Couoh-Cardel et al., 2015*; *Qi and Forgac, 2008*). In the structure of dissociated yeast $V_o$, the hydrophilic region of the *a* subunit ($a_{sol}$) changes its conformation to prevent rotation of the rotor complex (*Roh et al., 2018*; *Mazhab-Jafari et al., 2016*). The yeast $a_{sol}$ region lies in close proximity to the *d* subunit, the rotor region of the isolated yeast $V_o$ component. Both the $a_{sol}$ region and *d* subunit represent the trademarks of the V-ATPase family lacking in F-ATPases (see *Figure 1A–C*; *Iwata et al., 2004*). Thus, the $a_{sol}$ region and the *d* subunit appear to be crucial in stabilizing the auto-inhibition structure of the dissociated $V_o$ domain.

A regulatory dissociation/association mechanism has not been reported for the bacterial V/A-ATPase, however, reconstitution experiments suggest an assembly pathway for the *holo* complex, in which cytosolic $V_1$ associates with membrane $V_o$ (*Figure 1—figure supplement 1B*; *Kishikawa and Yokoyama, 2012*). Thus, proton leak through the $V_o$ domain in *Tth* membranes may also be blocked

by an autoinhibition mechanism similar to that in the eukaryotic enzyme. Indeed, the $Tth$ V/A-ATPase and eukaryotic V-ATPase share very similar structures with $V_o$ moieties comprising the $a$ and $d$ subunits in addition to the c ring.

Structural analysis of several subunits and subcomplexes of V/A-ATPases has been successfully carried out (*Iwata et al., 2004*; *Makyio et al., 2005*; *Toei et al., 2007*; *Nagamatsu et al., 2013*; *Murata et al., 2005*). Recent advances of single particle cryogenic microscopy (cryoEM) have facilitated structural analysis of the entire *holo* complexes of prokaryotic and eukaryotic V-ATPases in several rotational states (*Nakanishi et al., 2018*; *Zhou and Sazanov, 2019*; *Vasanthakumar et al., 2019*). While several structures of the isolated yeast $V_o$ have been reported (*Roh et al., 2018*; *Mazhab-Jafari et al., 2016*; *Vasanthakumar et al., 2019*), a high resolution structure of the isolated $Tth$ $V_o$ is currently unavailable, limiting understanding of the mechanism of enzyme inhibition.

Here, we report a cryoEM structure of isolated $Tth$ $V_o$ at 3.9 Å resolution. The $V_o$ structure reveals that the $a_{sol}$ region and $d$ subunit adopt distinct conformation, appropriate for inhibiting rotation of $d_1c_{12}$ relative to the stator $a$ subunit. This conformation is different from that seen in the $V_o$ moiety of the complete $Tth$ V/A-ATPase. Biochemical analysis using $Tth$ $V_o$ reconstituted into liposomes supports inhibition of proton conductance of isolated $V_o$ with a threshold membrane potential. Our results indicate that bacterial and eukaryotic $V_o$ domains use a similar mechanism for auto-inhibition of proton conductance. This mechanism prevents proton leak from $Tth$ cells through an intermediate assembly of the $V_o$ domain of $holo$ V/A-ATPase under physiological conditions.

## Results

### CryoEM structures of the isolated $V_o$ domain and *holo Tth* V/A-ATPase

We purified both the $Tth$ V/A-ATPase and $V_o$ domain with a His$_3$-tagged $c$ subunit from membranes of $T. thermophilus$ cells using Ni-NTA resin. The purified complexes were reconstituted into nanodiscs composed of the membrane scaffold protein MSP1E3D1 and POPC lipids. For the $Tth$ V/A-ATPase, acquisition of micrographs was carried out using the Titan Krios electron microscope equipped with a Falcon II direct electron detector. Cryo-EM micrographs of the complexes reconstituted into nanodiscs resulted in higher resolution EM maps compared to those previously reported for the LMNG solubilized preparations (*Nakanishi et al., 2018*). The strategy of single particle analysis for the $Tth$ V/A-ATPase is summarized in *Figure 2—figure supplement 1*. We reconstructed the 3D structure of the $holo$ complex rotational state 1 using 71,196 polished single particle images. The final structure of the state one has an overall resolution of 3.6 Å (*Figure 2A*). After subtracting the EM density of the membrane embedded domain from the density of the whole complex, we obtained a focused density map of $A_3B_3D_1F_1d_1$ with two EG peripheral stalks and the soluble arm domain of the a subunit ($a_{sol}$) at 3.5 Å resolution (*Figure 2—figure supplement 5*). This map allowed us to build an atomic model of $A_3B_3D_1F_1$ ($V_1$) (*Figure 2—figure supplement 6*). In our map, the obvious density of ADP-Mg was observed in the closed catalytic site, but not clearly observed in the semi-closed site, in contrast to our previously reported structure of the state 1 (PDBID: 5Y5Y). The secondary ADP in the semi-closed site shows a lower occupancy due to low affinity of the semi-closed site for the nucleotide and partial flexibility of the complex (*Figure 2—figure supplement 2A*). In the recent cryoEM map of $Tth$ V/A-ATPase (PDBID: 6QUM), clear densities, likely corresponding to ADP, were observed in the cavities of the crown-like structure formed by the six β barrel domains of $A_3B_3$ (*Zhou and Sazanov, 2019*). In contrast, these densities were not clearly visible in our structure (*Figure 2—figure supplement 2B*). This dissimilarity can presumably be explained by differences in the purification procedures; we purified the His-tagged $Tth$ V/A-ATPase using a nickel column, while the authors of the previous study isolated their $Tth$ V/A-ATPase without an affinity purification step.

The purified $V_o$ domain reconstituted into nanodiscs was subjected to single particle analysis using a cryoEM (CRYOARM200, JEOL) equipped with a K2 summit electron direct detector in electron counting mode. The strategy of single particle analysis for $TthV_o$ is summarized in *Figure 2—figure supplement 3*. The 2D class averages disclosed the isolated $V_o$ domain with clearly visible transmembrane helices and a hydrophilic domain extending above the integral membrane region (*Figure 2—figure supplement 4*). The scaffold proteins and lipids of the nanodiscs surrounding the

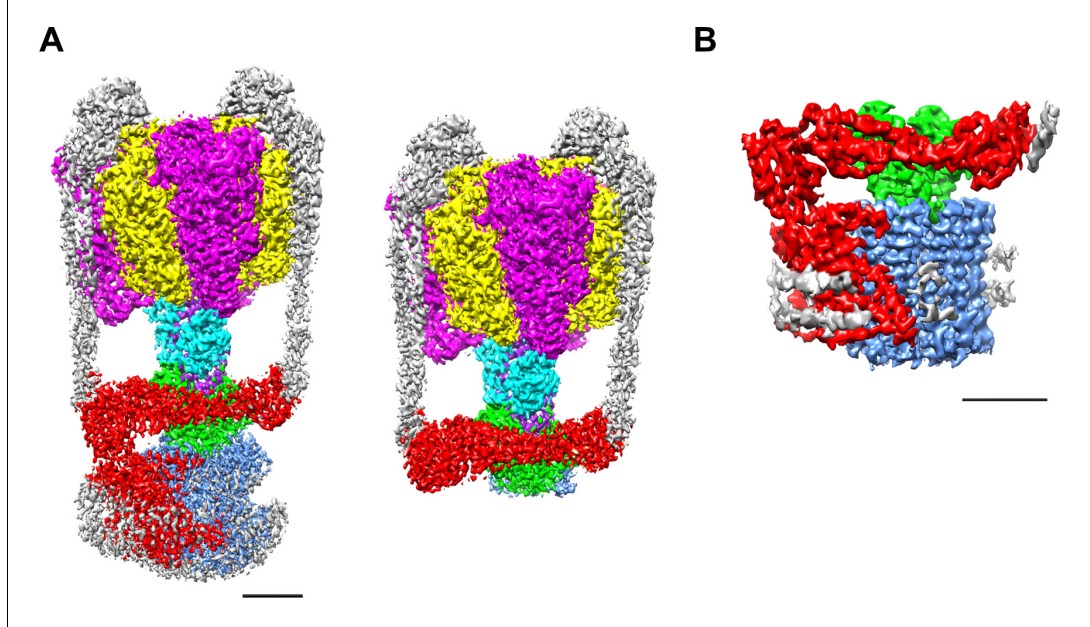

**Figure 2.** EM density map of the enzyme complex. (A) The *holo Tth* V/A-ATPase (left) and the focused refined map of $A_3B_3DFd(EG)_2a_{sol}$ (right). (B) The isolated $V_o$ domain. Densities corresponding to the individual subunits are colored as follows: A, magenta; B, yellow; D, purple; F, cyan; E and G, gray; *a*, red; *d*, green; and *c*, dark blue. Scale bar = 30 Å.

The online version of this article includes the following figure supplement(s) for figure 2:

**Figure supplement 1.** Single particle analysis of *Tth* V/A-ATPase.

**Figure supplement 2.** Structures of the nucleotide binding sites in the focused refined map of $A_3B_3DFd(EG)_2a_{sol}$.

**Figure supplement 3.** Single particle analysis of $Tth V_o$.

**Figure supplement 4.** 2D class averaged images of the isolated $V_o$ using polished particles.

**Figure supplement 5.** Local resolution maps of the *holo* $TthV/A$-ATPase (A) and isolated $V_o$ (B).

**Figure supplement 6.** Representative densities with model fitting for each subunit of $TthV_1$.

**Figure supplement 7.** Representative densities with model fitting for each subunit of $TthV_o$.

membrane domain of the isolated $V_o$ were clearly visible. Subsequent 3D classification of the observed $V_o$ states revealed only one major class, indicating that the isolated $V_o$ is structurally homogenous, in contrast to the *Tth* V/A-ATPase, which was clearly visible in three different rotational states (*Nakanishi et al., 2018*). Our 3D reconstruction map of the isolated $V_o$ complex was obtained with an overall resolution of 3.9 Å (*Figure 2—figure supplement 5*). The final map shows clear density for protein components of $V_o$, including subunit *a*, subunit *d* and the $c_{12}$ ring, but the EM density for both EG stalks, which attach to the $a_{sol}$ region, is weak indicating disorder or flexibility in these regions (*Figure 2B*). In this structure, a C-terminal region of the EG stalk on the distal side is visible. With the exception of these two EG stalks, side-chain densities are detectable for most of the proteins in the complex, allowing construction of a de novo atomic model using Phenix and Coot software (*Figure 3A,B*, *Figure 2—figure supplement 7*). The map contains an apparent density inside the $c_{12}$ rotor ring, likely corresponding to the phospholipids capping the hole of the ring (*Figure 3—figure supplement 1A*). A further apparent density was identified in the cavity between the *a* subunit and $c_{12}$ ring on the upper periplasmic side (*Figure 3—figure supplement 1B*). This density may also correspond to phospholipids, and we suppose that it functions to plug the cavity between the *a* subunit and $c_{12}$ ring, preventing proton leak from the periplasmic proton pathway. Similar densities corresponding to phospholipids were also observed in the recently published cryoEM density map of the *holo* complex (*Zhou and Sazanov, 2019*). Notably, the diameter of the $c_{12}$ rotor ring in the isolated $V_o$ is slightly smaller than that in the *Tth* V/A-ATPase (*Figure 3—figure supplement 2*). It is likely that penetration of the short helix of the subunit D into the subunit cavity of subunit *d* enlarges the diameter of the $c_{12}$ rotor ring in the *Tth* V/A-ATPase.

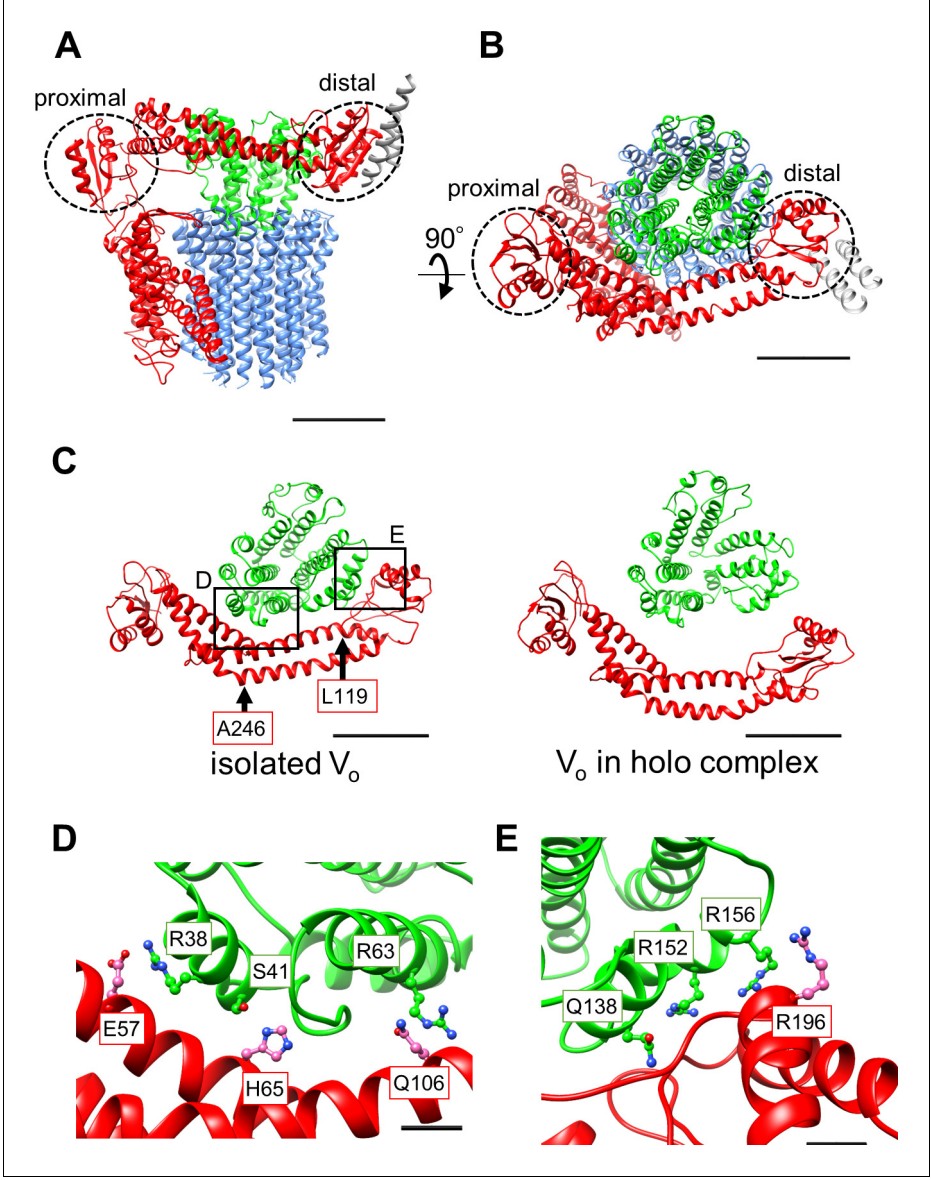

**Figure 3.** Atomic model of the isolated $V_o$ domain. (A) Side view and (B) top view of *a*-, *d*-, *c*-, and EG subunits colored as in *Figure 2*, respectively. Scale bar represents 30 Å. The proximal and distal subdomains of the *a*-subunit are circled by dotted lines. (C) Comparison of the relative positions of $a_{sol}$ (red) and the d subunit (green) in the isolated $V_o$ domain (left) and the $V_o$ domain in the *holo* complex (right). Arrows indicate the kinking and twisting points in the $a_{sol}$ region of the isolated $V_o$. Scale bar represents 30 Å. (D) and (E) Specific interactions between the $a_{sol}$ region and *d* subunit at the proximal (D) and distal (E) regions. The regions are specified by black squares in C. Scale bar = 5 Å.

The online version of this article includes the following figure supplement(s) for figure 3:

**Figure supplement 1.** The density corresponding to lipids in *Tth* $V_o$.
**Figure supplement 2.** Comparison of circumferences of the rotor *c*-ring in the isolated $V_o$ and *holo*-complex.

## Structure comparison of the isolated and complexed $V_o$ domains

A comparison of structures determined for the isolated $V_o$ domain with that in the *holo* complex revealed a high degree of similarity in the membrane embedded region. However, there were significant differences in the *a* subunit. The basic structure of the *Tth* $V_o$ *a* subunit is almost identical to the eukaryotic counterpart, comprising a soluble arm domain ($a_{sol}$) and a C-terminal hydrophobic domain responsible for proton translocation via rotation of the $c_{12}$ ring. The $a_{sol}$ region contains two

globular $\alpha/\beta$ folding subdomains responsible for binding of both the proximal and distal EG stalks (*Figure 3A and B*). Both globular subdomains are connected by a hydrophilic coiled coil with a bent conformation.

In contrast to the $V_o$ structure in the *holo* complex, the $a_{sol}$ region in the isolated $V_o$ is located in close proximity to the *d* subunit as a result of kinking and twisting of the coiled coil at residues *a*/L119 and *a*/A246 (*Figure 3C*, indicated by the arrows). In this structure, several interactions between the *d* subunit and $a_{sol}$ residues can be observed (*Figure 3D*). At the proximal site, three amino acid residues, *a*/E57, *a*/H65, and *a*/Q106, form salt bridges or hydrogen bonds with residues *d*/R38, *d*/S41, and *d*/R63 of the *d* subunit, respectively. Our structure also reveals clearly connected densities between the distal subdomain of the $a_{sol}$ region and *d* subunit (*Figure 3E*). Four side chains, *d*/Q138, *d*/R152, *d*/R156, and *a*/R196 apparently form hydrogen bonds with the oxygen atoms in the main chain of *a*/E201, *a*/L144, *a*/A197, and *d*/R156, respectively. With the exception of the interaction between *a*/E57 and *d*/R38 in the proximal site, these interactions are broken by the dynamic movement of $a_{sol}$ and conformational changes of the *d* subunit in the $V_o$ moiety of the *holo Tth* V/A-ATPase. The conformational changes induced by binding of $V_1$ ($A_3B_3DF$) to $V_o$ are described in a separate section below.

## Structure of the membrane embedded region of the isolated $V_o$ domain

Our previous low-resolution structure of the *Tth* V/A-ATPase suggested the involvement of half-channels in proton translocation on both the cytoplasmic and periplasmic sides of the $V_o$ domain (*Nakanishi et al., 2018*). The atomic model of $V_o$ presented here reveals details of the half-channels formed by the membrane-embedded C-terminal region of the *a* subunit ($a_{CT}$) and its interface with the $c_{12}$ ring. The $a_{CT}$ region contains eight membrane-embedded helices, MH1 to MH8. MH7 and MH8 are the highly-tilted membrane-embedded helices characteristic of rotary ATPases. The cytoplasmic hydrophilic cavity is formed by the cytoplasmic side of MH4, MH5, MH7, and MH8, and the *c* subunit/chain Z. The cavity is lined by the polar residues, *a*/R482, *a*/H491, *a*/H494, *a*/E497, *a*/Y501, *a*/E550, *a*/Q554, *a*/T553, *a*/H557, and *c*(Z)/Thr54 (*Figure 4A*), which make up the cytoplasmic half-channel. The periplasmic sides of MH1, MH2, MH7, and MH8 form the periplasmic hydrophilic cavity, lined with *a*/D365, *a*/Y368, *a*/E426, *a*/H452, *a*/R453, *a*/D455, and *c*(Y)/E63. The two hydrophilic channels are separated by a salt bridge formed between *c*(Z)/63Glu, a residue critical for proton translocation, and *a*/Arg563, *a*/Arg622, *a*/Gln619 of MH7 (*Figure 4B*). This salt bridge is conserved in both eukaryotic and prokaryotic $V_o$ (*Mazhab-Jafari et al., 2016*; *Kishikawa and Yokoyama, 2012*). Of note, a salt bridge forms between a single arginine residue and a single glutamic (or aspartic) acid residue in $F_o$ (*Kühlbrandt, 2019*; *Murphy et al., 2019*; *Guo et al., 2019*). Similar to the two-channel model described for other rotary ATPases (*Srivastava et al., 2018*; *Hahn et al., 2018*), the two arginine residues on the MH7 and MH8 play an important role in protonation and deprotonation of the carboxy groups on the $c_{12}$ ring, with the resulting rotation of $dc_{12}$ driven by proton translocation from the periplasmic to cytoplasmic side (*Guo and Rubinstein, 2018*; *Hahn et al., 2018*; *Pogoryelov et al., 2010*). Notably, in addition to the rigid salt bridge formed between the two *a*/Arg residues, *a*/Gln and *c*/Glu, further interactions between $a_{ct}$ and the $c_{12}$ ring are observed. Furthermore, *a*/Asp392 and Leu393 -*c*(Y)/Arg49 in the loop region of the *c* subunit (*Figure 4—figure supplement 1A*), and the periplasmic sides of MH5 and MH6 are in close proximity to the C-terminal end of the *c* subunit (*Figure 4—figure supplement 1B*). These interactions are observed in the $V_o$ moiety of the recently published *holo* complex structure (*Zhou and Sazanov, 2019*). Overall, our $V_o$ structure is largely identical to the $V_o$ moiety observed in the *holo* complex with the exception of some alterations in the hydrophilic domain (*Zhou and Sazanov, 2019*).

## Voltage threshold for proton conductance activity of the isolated $V_o$ domain

Our structure of the isolated $V_o$ domain suggests that the rotation of the $c_{12}$ rotor ring relative to the stator is mechanically hindered by a defined interaction between the $a_{sol}$ region and the *d* subunit. Previous studies have shown that isolated yeast $V_o$ is impermeable to protons (*Couoh-Cardel et al., 2015*; *Qi and Forgac, 2008*), but it was unclear whether proton conductance is also

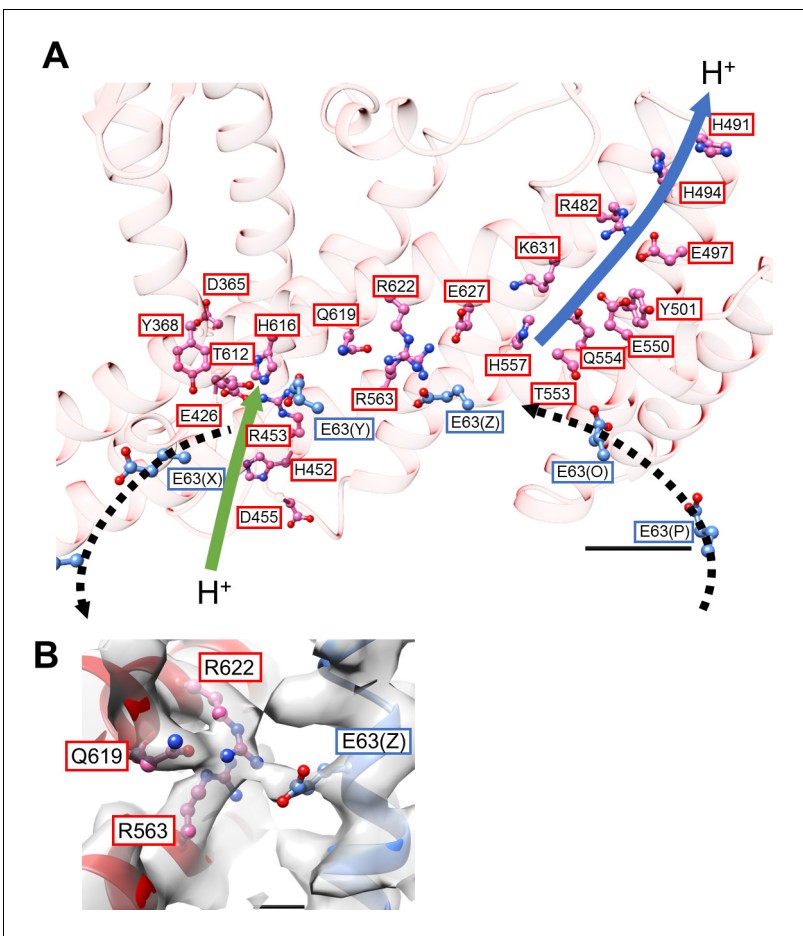

**Figure 4.** Structure of the hydrophobic domain of isolated $V_o$. (**A**) The half-channels for proton translocation on both the cytoplasmic and periplasmic sides of the isolated $V_o$ domain. Residues lining the pathways are represented as balls and sticks. Residues from the *a*-subunit and *c*-subunit are indicated in red and blue boxes, respectively. Proton flow, as it would occur in the case of ATP synthesis, is represented by arrows. The solid arrows indicate proton flow from the periplasmic side to the *c*-subunit (green), and from the *c*-subunit to the cytoplasmic side (blue). The dotted black arrows indicate proton movement due to rotation of the $c_{12}$-ring. Scale bar = 10 Å. (**B**) The salt bridge between *a*/Arg563, Arg622, Gln619 and *c*/Glu63. Scale bar = 3 Å.

The online version of this article includes the following figure supplement(s) for figure 4:

**Figure supplement 1.** Interactions between the *a*-subunit and $c_{12}$-ring.

inhibited in the isolated *Tth* $V_o$ domain. To investigate proton conductance through the isolated *Tth* $V_o$, we reconstituted this domain into liposomes energized with a $\Delta\psi$ generated through a potassium ion ($K^+$)/valinomycin diffusion potential. The pH change in the liposomes was monitored with 9-Amino-6-Chloro-2-Methoxyacridine (ACMA); the emission traces at 510 nm excited at 460 nm were recorded (*Figure 5*). The membrane potential was modulated by varying the external $K^+$ concentration according to the Nernst equation. As shown in *Figure 5B*, the isolated $V_o$ domain displays no proton conductance when the membrane potential is lower than 120 mV, defining a voltage threshold. The proton conductance through the $V_o$ increases proportionally with the membrane potential when the membrane potential exceeds 130 mV (*Figure 5B*). The reported membrane potential in bacterial cells varies from −75 to −220 mV depending on growth environment and method of quantification (*Lo et al., 2007*; *Bot and Prodan, 2010*). Although the membrane potential of *T. thermophilus* under physiological conditions is unknown, we reported previously that the *Tth* V/A-ATPase is capable of ATP synthesis when the membrane potential exceeds −110 mV (*Toei et al., 2007*). Thus, proton impermeability of the isolated *Tth* $V_o$ observed at potentials less than −120 mV may function to maintain *pmf* for ATP synthesis, when *Tth* $V_o$ exists solely on the cell membrane. In contrast to the

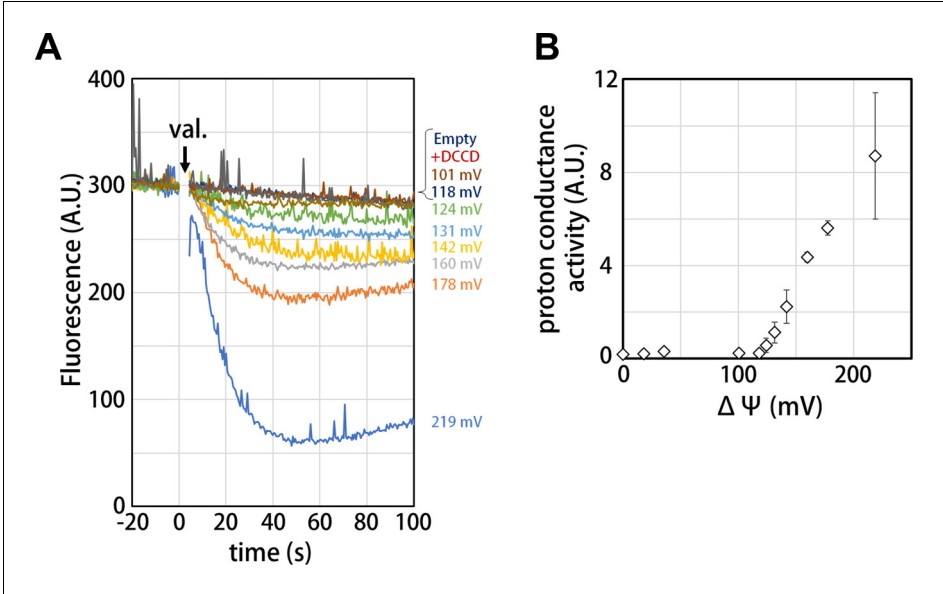

**Figure 5.** Proton conductance through the isolated $V_o$ domain. (**A**) Changes of ACMA fluorescence due to pH changes inside the $V_o$ proteo-liposomes. Values of the membrane potential ($\Delta\Psi$) were estimated using the Nernst equation, $\Delta\Psi = RF/zF \ln[KCl]_o/[KCl]_i$, as described in the Materials and methods section. (**B**) The voltage threshold of proton conductance through the $V_o$ domain (mean ± SD, n = 3).

$V_o$ domain, several experiments have indicated that proton conductance through the bacterial $F_o$ domain is not sensitive to any specific threshold in membrane potential (*Wiedenmann et al., 2008*), whereas bacterial $F_oF_1$ is sensitive to a membrane potential threshold, likely dependent on the interaction between $F_o$ and $F_1$ (*Feniouk et al., 2004*). In addition, proton conductance through the $F_o$ domain increases linearly with increasing $\Delta\psi$ loaded on the $F_o$ liposome. These results indicate that there are no or few interactions between the *a* subunit and *c*-ring to hinder *c*-ring rotation in $F_o$. Together, the observed results suggest that $a_{sol}$ of the *a* subunit and the *d* subunit, absent from $F_o$ and validated structures of the V type ATPases, can be one of the keys for mechanical inhibition of proton conductance through $V_o$.

## Discussion

The structure of the isolated *Tth* $V_o$ obtained clearly shows a different conformation from the $V_o$ moiety in the holo-complex. From structural comparison between isolated $V_o$ and the *holo* complex, it can be suggested that structural changes in isolated $V_o$ observed in two subunits were most likely induced by dissociation of the $V_1$ domain from $V_o$. In the isolated $V_o$ domain, the *d* subunit adopts the closed form in which three side chains of the *d* subunit are able to interact with the distal subdomain of $a_{sol}$ (*Figure 3E*). Once the short helix of the D subunit, an axis subunit of the $V_1$ domain, inserts into the cavity of the *d* subunit, the interaction between H6 and H11 via *d*/R90 and *d*/E195 is broken (*Figure 6A* and *Video 1*), resulting in the *d* subunit adopting an open form, with side chains orientated away from the distal subdomain of $a_{sol}$.

Another contributing factor is the dynamic motion of the $a_{sol}$ region induced by binding the distal EG stalk to the top of the $A_3B_3$ from the $V_1$ domain. In the isolated $V_o$, the N-terminal region of the EG stalk bound to the distal subdomain of $a_{sol}$ is at a much steeper angle relative to the horizontal coiled coil structure of $a_{sol}$ than that in the *holo* enzyme (*Figure 6B,C* and *Figure 6—figure supplement 1*). This finding suggests that the stalk region also adopts a steep angle, although the stalk and head domain of EG are disordered in the resolved structure and thus not visible in the density map (*Figure 2B*). Once the C-terminal globular domain of the distal EG stalk binds onto the top of $A_3B_3$, the angled distal EG adopts a vertical standing form, resulting in both twisting and kinking of the coiled coil of the hydrophilic arm and the distal globular subdomain of the *a* subunit (*Figure 6C*,

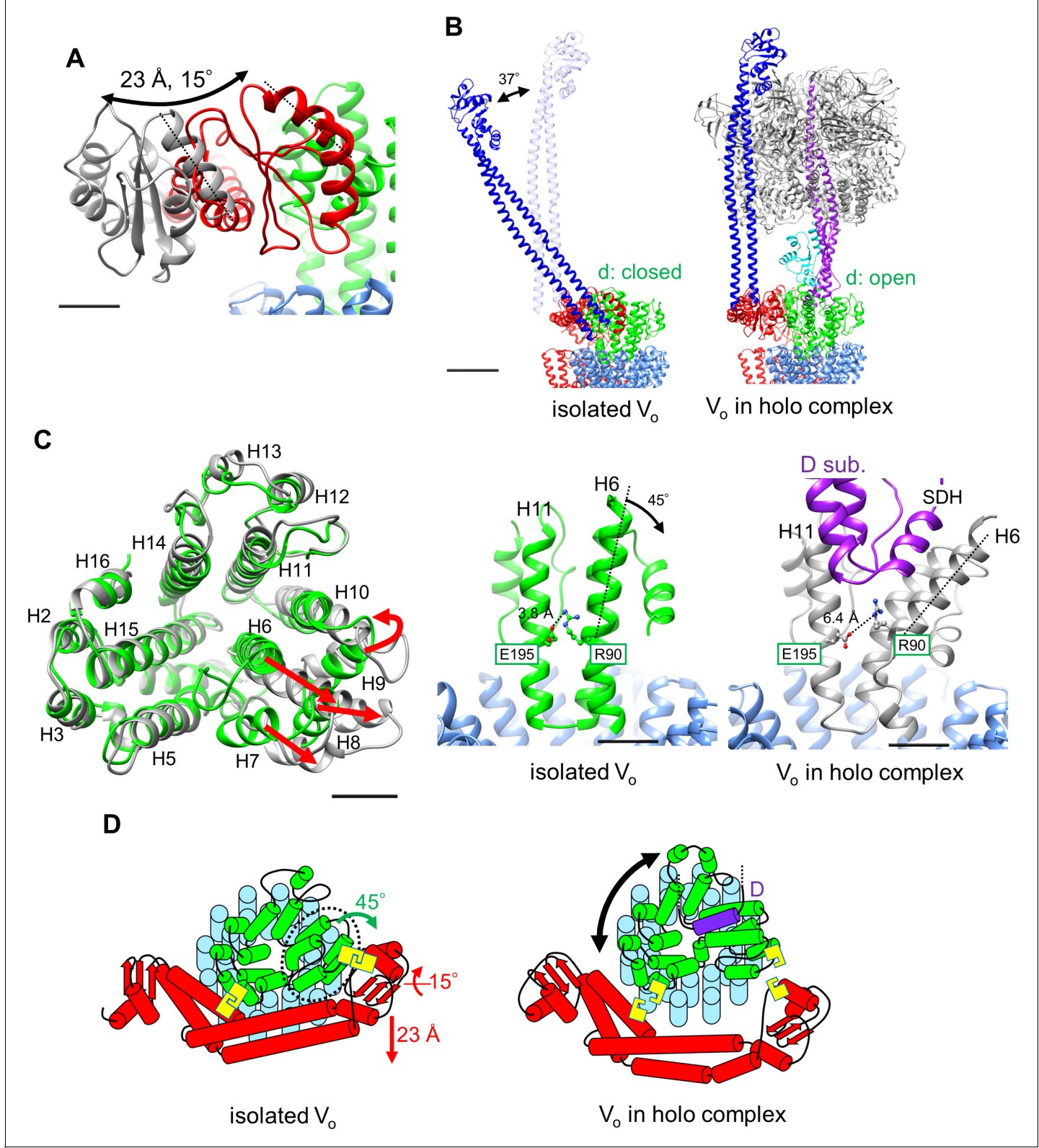

**Figure 6.** Conformational changes occurring in both the *d*- and *a*$_{sol}$ subunits as a result of $V_1$ to $V_o$ binding. (**A**) Structural changes in the *d* subunit caused by insertion of the screw driver helix (SDH). A top view of the *d*-subunit is shown in the left panel. The *d*-subunit from the isolated $V_o$ domain and the *holo* enzyme are colored in green and grey, respectively. Red arrows indicate movements of helices 6–9 (H6–H9). The key helices, H6 and H11, of the *d*-subunit in the isolated $V_o$ domain and *holo* complex are shown in panel A, center and right. The H6 helix bends 45° as a result of interaction

*Figure 6 continued on next page*

*Figure 6 continued*

between the d-subunit and SDH. (**B**) Structural changes in the distal subdomain of $a_{sol}$. Upon the pivoting movement of $a_{sol}$ on the proximal subdomain, the distal subdomain swings 25 Å and twists 15° between the isolated $V_o$ (red) and the *holo* complex (gray). (**C**) The EG structure in the distal subdomain of $a_{sol}$ ($EG_d$) in the isolated $V_o$ domain (left) and in the *holo* complex (right). (**D**) A schematic representation of mechanical inhibition of $V_o$ induced by dissociation of $V_1$. In the isolated $V_o$ domain, rotation of the central rotor is inhibited by interactions between the d-subunit and $a_{sol}$ (yellow box, *Figure 3D, E*).

The online version of this article includes the following figure supplement(s) for figure 6:

**Figure supplement 1.** Low resolution electron density map of isolated $V_o$.

**Figure supplement 2.** Conformational changes in the *d* subunit induced by association of $V_1$ with $V_o$ in yeast V-ATPase (left) and in *Tth*V/A-ATPase (right).

*Video 2*). These dynamic motions of $a_{sol}$ induce disruption of specific interactions between $a_{sol}$ and the *d* subunit.

The isolated yeast $V_o$ domain also adopts a conformation where the $a_{sol}$ region is in close proximity to the *d* subunit, resulting in rigid interaction between the stator and rotor that is advantageous for inhibition of proton conductance (*Roh et al., 2018*; *Mazhab-Jafari et al., 2016*). Although an atomic model of the yeast *holo* V-ATPase has yet to be determined, a poly alanine model of the yeast V-ATPase shows that the $a_{sol}$ region is some distance away from the *d* subunit in the $V_o$ moiety (*Zhao et al., 2015*). In addition, a recently reported structure of the mammal V-ATPase clearly shows that $a_{sol}$ is at a distance where it cannot interact with the *d* subunit (*Abbas et al., 2020*). This structure suggests that a similar conformational change in $V_o$ is induced by binding of the $V_1$ domain in the yeast V-ATPase, as described by Oot and Wilkins previously (*Oot and Wilkens, 2012*). Notably, the *d* subunit in the yeast enzyme differ in conformation between the isolated $V_o$ domain and *holo* enzyme, in contrast to the *Tth* enzyme, where the *d* subunit is in the closed form in the isolated $V_o$ domain (*Figure 6—figure supplement 2*; *Vasanthakumar et al., 2019*). The *d* subunit from the mammalian *holo* V-ATPase adopts a more open conformation than the yeast *d* subunit from the isolated $V_o$ complex, as seen in the *holo Tth* V/A-ATPase (*Abbas et al., 2020*). In addition, Abbas et al. suggest that the *d* subunit from the yeast *holo* V-ATPase is also more open compare to that of the yeast isolated $V_o$ (*Abbas et al., 2020*). These results indicate that the *d* subunit in the mammalian and yeast V-ATPase also exhibits a conformational change between isolated $V_o$ and *holo* enzyme.

However, Qi et al. reported that yeast $V_o$ was impermeable to proton even in the absence of interactions between the *a*-subunit and *d*-subunit (*Couoh-Cardel et al., 2015*; *Qi and Forgac, 2008*). These findings suggest that the interactions between $a_{sol}$ and *d*-subunit are not the only mechanism by which proton permeability is inhibited. In fact, salt bridges between the arginine residues (*a*/R563, R622 in *Tth* $V_o$, *a*/R735, 799 in yeast $V_o$) and the glutamate residue (*c*/E63 in *Tth* $V_o$, *c*/E108 in yeast $V_o$) (*Roh et al., 2018*; *Mazhab-Jafari et al., 2016*; *Vasanthakumar et al., 2019*) are identified in isolated $V_o$ from both *T. thermophilus* and yeast $V_o$. These salt bridges between the stator *a* subunit and the rotor *c*-ring inhibit proton permeability by hindering *c*-ring rotation (*Mazhab-Jafari et al., 2016*). It is still

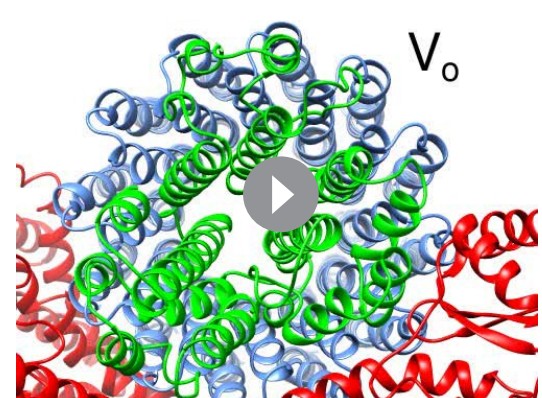

**Video 1.** Conformational changes of the *d*-subunit between isolated $V_o$ and $V_o$ in holo-enzyme. A morphed movie focuses on the conformational changes of the *d*-subunit between isolated $V_o$ and $V_o$ in the holo-enzyme. *a*-, *d*-, *c*- and *d*-subunits are colored in red, green, dark blue, and orange, respectively. *d*/R90 and *d*/E195 are represented as balls and sticks. The density map of isolated $V_o$ is shown as a semi-transparent surface.
https://elifesciences.org/articles/56862#video1

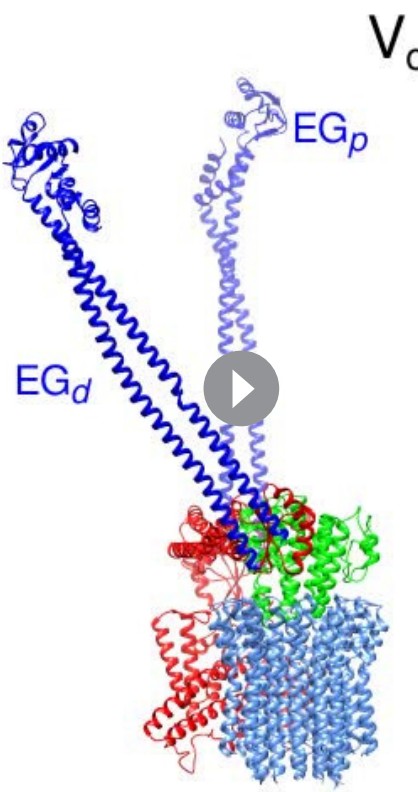

**Video 2.** Conformational changes of a hydrophilic arm of the *a*-subunit linked to movement of the EG subunits. A morphed movie focusing on the conformational changes of a hydrophilic arm of the *a*-subunit between isolated $V_o$ and $V_o$ in the holo-enzyme. $EG_p$ and $EG_d$ indicate proximal and distal EG subunits. The position of $EG_d$ was determined from the low-resolution density map (*Figure 6—figure supplement 1*). The hydrophilic arm of the *a*-subunit associated with binding $EG_d$ to one of the B subunits from $V_1$, is forced to swing away from the *d*-subunit, resulting in disruption of the specific interaction between the subunits.

https://elifesciences.org/articles/56862#video2

controversial whether the formation of this salt bridge represents a *bona fide* process of proton translocation that links deprotonation and re-protonation of glutamate residues in the *c* subunits. (*Kühlbrandt, 2019*; *Roh et al., 2018*; *Zhou and Sazanov, 2019*; *Abbas et al., 2020*; *Krah et al., 2019*; *Pierson et al., 2018*; *Symersky et al., 2012*). Undoubtedly, the salt bridge must be broken by both rotation of the *c*-ring driven by *pmf* and ATP hydrolysis in $V_1$ in order to perform the functions of ATP synthesis or proton pumping (*Figure 5*; *Roh et al., 2018*; *Vasanthakumar et al., 2019*; *Abbas et al., 2020*).

As described above, eukaryotic and prokaryotic V/A-ATPases appear to share a similar mechanism of conformational change at the $V_o$ moiety, advantageous for preventing proton leakage from cells or acidic vesicles. Nevertheless, there exist some interactions unique to *Tth* $V_o$, as described in this paper (*Figure 4—figure supplement 1*), and to yeast $V_o$, as reported by previous studies (*Roh et al., 2018*). This suggests that the auto-inhibition mechanisms of $V_o$ have been conserved during evolution of V type ATPases.

In the isolated yeast $V_o$, the luminal half-channel, which releases translocated protons to the lumen of acidic vesicles is closed and it is assumed to open transiently during catalysis (*Mazhab-Jafari et al., 2016*; *Krah et al., 2019*). In the case of *Tth* V/A-ATPase, both sides of the half-channel are open (*Zhou and Sazanov, 2019*). The membrane domain of the *a* subunit from the isolated *Tth* $V_o$ is largely identical to that of the *holo* enzyme (r. m. s. d. = 0.82 Å for A327-E637 of the *a* subunit), thus the half-channels are likely also open in *Tth* $V_o$ as observed in the *holo* enzyme. This indicates that *Tth*$V_o$ is more proton permeable than yeast $V_o$. This difference might be a consequence of the differences between the protein acting as an ATP synthase (*Tth* enzyme) and ATP driven proton pump (yeast enzyme), as previously suggested (*Krah et al., 2019*). Further studies, such as computational MD simulation, are required to assess the extent of contribution of each interaction to the auto-inhibition mechanism of *Tth* $V_o$.

Our structure of the isolated $V_o$ domain further reveals the mechanism of mechanical inhibition of rotation of the $dc_{12}$ rotor complex due to strong interactions between $a_{sol}$ and the *d* subunit, regions unique to V-ATPases. These interactions stabilize the isolated $V_o$ domain and protect against loss of the *d*-subunit in the absence of the rotor-stator interactions mediated by $V_1$ in the *holo* enzyme (*Ediger et al., 2009*). This stabilization of $V_o$ is likely to be a key factor for both assembly of *holo* V-type ATPase complexes and regulation of the eukaryotic V-ATPase via dissociation of $V_1$ from $V_o$.

# Materials and methods

## Key resources table

| Reagent type (species) or resource | Designation | Source or reference | Identifiers | Additional information |
|---|---|---|---|---|
| Strain, strain background *T. thermophilus* | HB8 | *Tamakoshi et al., 1997* | | |
| Chemical compound, drug | 14:0 PC (DMPC) | Avanti polar lipid | 850345 | |
| Chemical compound, drug | 16:0-18:1 PC (POPC) | Avanti polar lipid | 850457 | |
| Chemical compound, drug | n-Dodecyl-beta-D-maltopyranoside | cosmo bio | D-1304 | |
| Chemical compound, drug | Biobeads SM-2 | bio-rad | 1523920 | |
| Chemical compound, drug | L-α-Phosphatidylcholine from soybean, Type II-S | Merck | P5638 | |
| Chemical compound, drug | n-octyl-β-D-glucoside | sigma aldrich | 850511P | |
| Chemical compound, drug | 9-Amino-6-Chloro-2-Methoxyacridine | Thermo Fisher | A1324 | |
| Chemical compound, drug | Carbonyl cyanide 4-(trifluoromethoxy) phenylhydrazone | sigma aldrich | C2920 | |
| Chemical compound, drug | Valinomycin | sigma aldrich | V0627 | |
| Software, algorithm | RELION | *Zivanov et al., 2018* | RRID:SCR_016274 | |
| Software, algorithm | MotionCor2 | *Zheng et al., 2017* | RRID:SCR_016499 | |
| Software, algorithm | Gctf | *Zhang, 2016* | RRID:SCR_016500 | |
| Software, algorithm | COOT | http://www2.mrc-lmb.cam.ac.uk/personal/pemsley/coot/ | RRID:SCR_014222 | |
| Software, algorithm | Phenix | https://www.phenix-online.org/ | RRID:SCR_014224 | |
| Software, algorithm | MolProbity | http://molprobity.biochem.duke.edu | RRID:SCR_014226 | |

## Protein preparation

*T. thermophilus* V/A-ATPase was expressed with a His$_3$ tag on the C-terminus of the c-subunit using a modified operon generated by the integration vector system (*Tamakoshi et al., 1997*). Purification of His-tagged V$_o$ was carried out as described previously (*Nakanishi et al., 2015*). Briefly, membranes of *T. thermophilus* were suspended in a buffer containing 10% Triton X-100 and sonicated to solubilize membrane proteins. After ultracentrifugation, the supernatant containing V/A-ATPase was applied to a Ni-NTA column. The fractions containing V/A-ATPase were dialyzed against 20 mM Tris-HCl (pH 8.0), 1 mM EDTA for 2 days at 4°C. The combined fraction was applied to a Resource Q column. Eluted fractions were analyzed by SDS-PAGE and the fractions containing V$_o$ and V/A-ATPase were concentrated separately using Amicon 100K molecular weight cut-off filters (Millipore).

For nanodisc incorporation, 25 mM DMPC (Avanti) solubilized in 5% DDM was used. The concentrated V$_o$ fraction, the scaffold protein MSP1E3D1, and DMPC were mixed in a 1:12:600 molar ratio and incubated for 30 min at room temperature. Then, 200 µL of Bio Beads SM-2 equilibrated with wash buffer (20 mM Tris-HCl, pH8.0, 150 mM NaCl) were added into 500 µL of the protein-lipid mixture. After 2 hr of incubation at 4°C with gentle stirring, an additional 300 µL of Bio Beads was added to the mixture prior to overnight incubation at 4°C to form nanodisc. The supernatant of the mixture containing the nanodisc-V$_o$ was applied to a Superdex200 Increase 10/300 column equilibrated in

wash buffer. Individual fractions were analyzed by SDS-PAGE and concentrated to ~4 mg/mL. The prepared nanodisc-$V_o$ was stored at 4°C and used for cryo-grid preparation within several days.

V/A-ATPase was reconstituted into lipid nanodiscs using the same protocol as that for $V_o$, except that 1-Palmitoyl-2-oleoyl-sn-glycero-3-phosphocholine (POPC, Avanti) was used as the lipid during reconstitution. Purified V/A-ATPase solubilized in 0.03% n-Dodecyl-β-D-maltoside (DDM) was mixed with the lipid stock and membrane scaffold protein MSP1E3D1 (Sigma) at a specific molar ratio of V/A-ATPase: MSP: POPC lipid = 1: 4: 520 and incubated on ice for 0.5 hr. 200 μl of Bio-beads SM-2 were added to initiate the reconstitution by removing detergents from the system and the mixture was incubated at 4°C for 3 hr with constant rotation. The bio-beads were removed and the nanodisc mixture applied to a Superdex200 Increase 10/300 column (GE Healthcare) pre-equilibrated in buffer (20 mM Tris-HCl pH8.0, 150 mM NaCl, 2 mM $MgCl_2$). Reconstitution was assessed by both size exclusion chromatography and SDS-PAGE. The peak corresponding to the nanodisc-reconstituted V/A-ATPase was immediately used for cryo-EM observation.

## Biochemical analysis

For measurements of proton channel activity, purified Vo was reconstituted into liposomes. L-α-Phosphatidylcholine Type II-S (Sigma-Adrich) was washed repeatedly beforehand to eliminate contamination of $K^+$ ions (*Soga et al., 2012*), and the L-α-Phosphatidylcholine suspension was adjusted to a final concentration of 40 mg/mL in 4 mM Tricin, and 5 mM $MgCl_2$ . To 250 μL of L-α-Phosphatidylcholine suspension, 250 μL of a solution containing 8% (w/v) n-octyl-β-D-glucoside (Sigma), and 500 mM KCl were added. Then, 60–80 ng of purified $V_o$ was added. After a 30 min incubation at 4°C, 200 μL of Bio Beads SM-2, pre-equilibrated in 2 mM Tricin and 2.5 mM $MgCl_2$, were added to the mixture. The bead mixture was gently stirred for 30 min at room temperature. After that, 300 μL of Bio Beads were added to the mixture and incubated for another 2 hr. The supernatant was ultracentrifuged (40 k rpm, 4°C, 30 min) to remove contaminating KCl. The pellet containing reconstituted proteoliposome was re-suspended in 2 mM Tricin and 2.5 mM $MgCl_2$. The proteoliposomes were used for proton channel assay immediately. Proton channel activity was detected by the fluorescence quenching of 9-amino-6-chloro-2-methoxyacridine (ACMA) (Thermo Fisher), which changes fluorescence in response to pH reduction inside the proteoliposome. Fluorescence changes were monitored using a spectrofluorometer (FP-6200, JASCO). A 1200 μL aliquot of reaction buffer (2 mM Tricin, pH8.0, 2.5 mM $MgCl_2$, 500 mM KCl + NaCl, 1 μL of 30 mg/mL ACMA, 20 μL of proteoliposome) were incubated at 25°C. Proton channel activity was initiated by injection of 1 μL of 0.1 mg/mL valinomycin at the time = 50 s. After 100 s, 1 μL of 0.2 mg/mL carbonyl cyanide-p-trifluoromethoxyphenylhydrazone was added. The initial rate of pH change was estimated from the linear fitting of the initial decay of fluorescence. The membrane potential ($\Delta\Psi$) across the liposome membrane was calculated by the Nernst equation, $\Delta\Psi = (kBT/zF)\ln([K^+]_{out}/[K^+]_{in})=59.2 \log([K^+]_{out}/[K^+]_{in})$ in mV at 25°C, where $[K^+]_{out}$ was taken to be that of the reaction buffer, and $[K^+]_{in}$ was 500 mM as in the buffer for proteoliopsome reconstitution.

Protein concentrations of $V_o$ were determined from UV absorbances calibrated by quantitative amino acid analysis; 1 mg/ml gave an optical density of 0.56 at 280 nm. Polyacrylamide gel electrophoresis in the presence of SDS or AES was carried out as described previously (*Nakano et al., 2008*). The proteins were stained with Coomassie Brilliant Blue.

## EM imaging

For cryo-grid preparation, Quanfifoil R1.2/1.3 molybdenum grids were glow discharged by an Ion Bombarder (Vacuum Device) for 1 min. 2.4–2.7 μL of nanodisc-$V_o$ were loaded onto the grid and blotted for 9 s with a blot force of 10, wait time of 0 s at 4°C, and 100% humidity using a Vitrobot (FEI). Then, the grid was plunged into liquid ethane without drain time. Cryo-EM movie collection was performed with the CRYOARM200 (JEOL) operating at 200 keV accelerating voltage and equipped with a direct electron detector, K2 Summit (Gatan) in electron counting mode using the data collection software JADAS. The pixel size was 1.1 Å/pix (x5,0000), a total dose of 79.2 e⁻/ $Å^2$ (1.32 e⁻/ $Å^2$/frame) with a 12 s exposure time (60 frames), and a defocus range of −1.0 to −3.5 μm.

For V/A-ATPase analysis, gold grids were used to reduce beam-induced movement (*Russo and Passmore, 2014*). A 2.4 μL aliquot of V/A-ATPase sample at 3.5 mg/ml was added to a 1.2 μm hole, 1.3 μm spacing holey gold grid (Quantifoil UltrAuFoil) in a semi-automated vitrification device

(Vitrobot, FEI/Thermo Fisher) at 100% humidity, 4°C. The grid was then automatically blotted once from both sides with filter paper for a 9 s blot time. The grid was then plunged into liquid ethane without a delay time. Preparations of the V/A-ATPase were observed with a Titan Krios (FEI/Thermo Fisher) operating at 300 kV acceleration voltage and equipped with a Falcon II (FEI/Thermo Fisher) detector at a magnification of 75,000x with a pixel size of 1.1 Å, set up to capture 34 frames, corresponding to a total dose of 91 e⁻/ $Å^{-2}$ in a defocus range of −2.4 to −3.0 μm.

## Image processing

Image processing was performed using the Relion 3.0.7 software (*Zivanov et al., 2018*). A total of 5988 cryo-EM movies were collected for isolated $V_o$ and 3694 movies collected for V/A-ATPase. All images were subjected to motion correction using the MotionCor2 program (*Zheng et al., 2017*) followed by contrast transfer function (CTF) estimation using Gctf (*Zhang, 2016*). Manual selection of the motion-corrected micrographs results in 3268 good isolated $V_o$ micrographs and 3084 good V/A-ATPase micrographs. For $V_o$, a template for particle auto-picking was generated by 2D classification of particles picked by the LoG (Laplacian of Gaussian) method implemented in the Relion software, while particles were picked manually to generate references for auto-picking for V/A-ATPase. $V_o$ and V/A-ATPase particles were picked from each selected micrograph by template-based auto-picking and classified by several rounds of reference-free 2D classification (3.14 × $10^6$ and 0.35 × $10^6$ particles images, respectively). After 2D classifications, 706,617 particles selected for $V_o$ and 147,292 particles selected for V/A-ATPase were subjected to several rounds of 3D classification, respectively. The initial model of $V_o$ was generated from the $V_o$ domain of our previous *T. thermophilus* V/A-ATPase structure (*Nakanishi et al., 2018*) using UCSF chimera (*Pettersen et al., 2004*). A total of 175,930 particles selected for $V_o$ and 71,196 particles selected for V/A-ATPase assigned into good 3D classes were subjected to 3D auto-refinement followed by CTF refinement of Bayesian polishing. Then, 157,618 $V_o$ particles were selected from the polished particles by 2D classification. Another round of 3D auto-refine, CTF refinement, and a final round of masked auto-refine gave a $V_o$ map at 3.9 Å resolution and a V/A-ATPase map at 3.6 Å resolution. The resolution was estimated based on the gold standard FSC = 0.143 criterion. However, while the membrane domain was visible it was not well refined in the V/A-ATPase map. This is likely to be due to the structural flexibility between the $V_o$ and $V_1$ domains in this class. Therefore, focused classification with signal subtraction of the membrane domain was carried out for the V/A-ATPase map to obtain high-quality maps and this gave a near-atomic resolution (3.5 Å resolution) map of the hydrophilic domain ($A_3B_3DFE_2G_2da_{sol}$).

## Model building

To generate an atomic model for the isolated $V_o$ domain, each subunit of the $V_o$ complex from the previous structure of the *T. thermophilus* V/A-ATPase (PDBID: 5Y5X) was fitted into the density map as a rigid body. Notably the a-subunit was divided into soluble and transmembrane domains and these domains fitted into the map separately. The rigid body structures were fitted against the density map manually using the COOT software (*Emsley et al., 2010*). Then, the manually fitted structures were refined using the phenix.real_space_refine program contained in the Phenix suite software (*Adams et al., 2010*). These processes were performed over several rounds. The geometry of the atomic model built in this study was checked using the MolProbity tool (*Table 1*; *Chen et al., 2010*).

Part of the 4.7 Å resolution hydrophilic domain structure of the *T. thermophilus* V/A-ATPase (PDBID: 5Y5Y) was fitted into the map of $A_3B_3DFE_2G_2da_{sol}$. A rough initial model was refined against the map with the Phenix suite phenix.real_space_refine program (*Adams et al., 2010*). The initial model was extensively manually corrected residue by residue in the COOT graphics program (*Emsley et al., 2010*), in particular with respect to side-chain conformations. The peripheral stalks and d-subunit were removed because of a low resolution in these regions. The corrected model was again refined by the phenix.real_space_refine program with secondary structure, and the resulting model manually checked by COOT (*Emsley et al., 2010*). This iterative process was performed for multiple rounds to correct any remaining errors until the model was in good agreement with the geometry, as reflected by the MolProbity score of 2.21 for isolated $V_o$ and 2.75 for $A_3B_3DFE_2G_2da_{sol}$.

**Table 1.** CryoEM data collection, refinement and model statistics.

| | TthV/A-ATPase | $V_1EGda_{sol}$ | Isolated $V_o$ |
|---|---|---|---|
| **Data collection** | | | |
| Electron microscope | Titan Krios | | CRYOARM200 |
| Electron detector | Falcon II | | K2 summit |
| Magnification | 75,000 | | 50,000 |
| Voltage (kV) | 300 | | 200 |
| Electron exposure ($e^-/\text{Å}^2$) | 91 | | 79.2 |
| Defocus range (μm) | 2.4–3.0 | | 1.0–3.0 |
| Pixel size (Å) | 1.1 | | 1.1 |
| Movie No. | 3694 | | 5988 |
| Frame per movie | 34 | | 60 |
| Automation software | EPU | | JADAS |
| **Data processing** | | | |
| Total extracted particles | $3.5 \times 10^5$ | | $3.14 \times 10^6$ |
| Total particle after 2D | 144,758 | | 706,617 |
| Resolution (Å) | 3.6 | 3.5 | 3,93 |
| Sharpening B-factor | −81.07 | −60.25 | −110.87 |
| EMDB ID | 30013 | 30014 | 30015 |
| **Model building and refinement** | | | |
| Initial models | - | 5Y5Y | 5Y5X, 1V9M |
| Building and refinement package | - | COOT, phenix | COOT, phenix |
| Total atom No. | - | 26,631 | 13,888 |
| Total residue No. | - | 3418 | 1894 |
| Total chain No. | - | 8 | 16 |
| Ligands | - | ADP | - |
| cc_mask | - | 0.85 | 0.82 |
| Ramachandran favored | - | 88.89% | 92.75% |
| Ramachandran outliers | - | 0.03% | 0.00% |
| Rotamer outliers | - | 11.45% | 0.37% |
| c-beta deviation | - | 0 | 0 |
| CaBLAM outliers | - | 5.94% | 2.84% |
| Clashscore | - | 6.38 | 20.01 |
| RMSD bonds (Å) | - | 0.006 | 0.006 |
| RMS angle (°) | - | 0.669 | 0.725 |
| MolProbity score | - | 2.73 | 2.26 |
| PDB ID | - | 6LY8 | 6LY9 |

For model validation against over-fitting, the built models were used for calculation of FSC curves against both half maps, and those were compared with the FSCs of the final models against the final density maps used for model building by the phenix.refine program .

## Acknowledgements

We are grateful to all the members of the Yokoyama Lab for their continuous support and technical assistance. Our research was supported by Grant-in-Aid for Scientific Research (JSPS KAKENHI), Grant Number 17H03648, and Takeda Science foundation to KY Our research was also supported by Platform Project for Supporting Drug Discovery and Life Science Research (Basis for Supporting

Innovative Drug Discovery and Life Science Research (BINDS)) from AMED under Grant Number JP17am0101001 (support number 1312), and Grants-in-Aid from 'Nanotechnology Platform' of the Ministry of Education, Culture, Sports, Science and Technology (MEXT) to KM (Project Number. 12024046). This work was also supported by JST CREST to K.M. (Grant Number. JPMJCR1865).

## Additional information

### Funding

| Funder | Grant reference number | Author |
|---|---|---|
| Japan Society for the Promotion of Science | 17H03648 | Ken Yokoyama |
| Japan Agency for Medical Research and Development | JP17am0101001 | Kaoru Mitsuoka |
| Ministry of Education, Culture, Sports, Science, and Technology | 12024046 | Kaoru Mitsuoka |
| Takeda Science Foundation | | Ken Yokoyama |
| Japan Science and Technology Agency | JPMJCR1865 | Kaoru Mitsuoka |

The funders had no role in study design, data collection and interpretation, or the decision to submit the work for publication.

### Author contributions

Jun-ichi Kishikawa, Data curation, Formal analysis, Validation, Investigation, Methodology, Writing - original draft, Writing - review and editing; Atsuko Nakanishi, Data curation, Formal analysis, Validation, Investigation, Methodology, Writing - original draft; Aya Furuta, Investigation, Methodology; Takayuki Kato, Resources, Data curation, Supervision, Writing - original draft; Keiichi Namba, Resources, Supervision; Masatada Tamakoshi, Resources; Kaoru Mitsuoka, Resources, Data curation, Supervision, Funding acquisition, Writing - original draft; Ken Yokoyama, Conceptualization, Supervision, Funding acquisition, Writing - original draft, Writing - review and editing

### Author ORCIDs

Jun-ichi Kishikawa https://orcid.org/0000-0003-3913-7330
Ken Yokoyama https://orcid.org/0000-0002-6813-1096

### Decision letter and Author response

Decision letter https://doi.org/10.7554/eLife.56862.sa1
Author response https://doi.org/10.7554/eLife.56862.sa2

## Additional files

### Supplementary files

• Transparent reporting form

### Data availability

The density maps and the built models for Tth VoV1, Tth V1 (focused refined), and Tth Vo were deposited in EMDB (EMDB ID; 30013, 30014, and 30015) and PDB (PDB ID; 6LY8 for V1 and 6LY9 for isolated Vo), respectively. All data is available in the main text or the supplementary materials.

The following datasets were generated:

| Author(s) | Year | Dataset title | Dataset URL | Database and Identifier |
|---|---|---|---|---|
| Kishikawa J, Naka- | 2020 | V/A-ATPase from Thermus | https://www.ebi.ac.uk/ | Electron Microscopy |

| | | | | |
|---|---|---|---|---|
| nishi A, Furuta A, Kato T, Namba K, Tamakoshi M, Mitsuoka K, Yokoyama K | | thermophilus | pdbe/entry/emdb/EMD-30013 | Data Bank, EMD-300 13 |
| Kishikawa J, Nakanishi A, Furuta A, Kato T, Namba K, Tamakoshi M, Mitsuoka K, Yokoyama K | 2020 | V1-ATPase built from cryo-EM map of V/A-ATPase from Thermus thermophilus. | https://www.ebi.ac.uk/pdbe/entry/emdb/EMD-30014 | Electron Microscopy Data Bank, 30014 |
| Kishikawa J, Nakanishi A, Furuta A, Kato T, Namba K, Tamakoshi M, Mitsuoka K, Yokoyama K | 2020 | The membrane-embedded Vo domain of V/A-ATPase from Thermus thermophilus | https://www.ebi.ac.uk/pdbe/entry/emdb/EMD-30015 | Electron Microscopy Data Bank, 30015 |
| Kishikawa J, Nakanishi A, Furuta A, Kato T, Namba K, Tamakoshi M, Mitsuoka K, Yokoyama K | 2020 | V/A-ATPase from Thermus thermophilus, the soluble domain, including V1, d, two EG stalks, and N-terminal domain of a-subunit. | https://www.rcsb.org/structure/6LY8 | RCSB Protein Data Bank, 6LY8 |
| Kishikawa J, Nakanishi A, Furuta A, Kato T, Namba K, Tamakoshi M, Mitsuoka K, Yokoyama K | 2020 | The membrane-embedded Vo domain of V/A-ATPase from Thermus thermophilus | https://www.rcsb.org/structure/6LY9 | RCSB Protein Data Bank, 6LY9 |

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
