## [Decision Letter]

**Acceptance summary:**

The work determined Cryo-EM structures of both the intact (holo) and membrane embedded portions of the V-ATPase from *Thermus thermophilus* and gives new insights and testable models for the structural and mechanism of the auto-inhibition of the complex, a long-standing open question.

**Decision letter after peer review:**

Thank you for submitting your article "The mechanical inhibition of the isolated V_o_ from V-ATPase for proton conductance" for consideration by *eLife*. Your article has been reviewed by two peer reviewers, and the evaluation has been overseen by a Reviewing Editor and Olga Boudker as the Senior Editor. The following individual involved in review of your submission has agreed to reveal their identity: Geoffry A. Davis (Reviewer #1).

The reviewers have discussed the reviews with one another and the Reviewing Editor has drafted this decision to help you prepare a revised submission.

Summary:

The work determined Cryo-EM structures of both the intact (holo) and membrane embedded portions of the V-ATPase from *Thermus thermophilus* in an attempt to understand the auto-inhibition of the membrane embedded V_0_. Analysis of structural differences between these forms enzyme support an interesting structural and functional mechanism for the auto-inhibition of V_0_.

Both reviews found the work to be exciting, but that some changes were needed. Their general consensus was that it should be possible to address these issues without additional experimental data, by deeper analyses of the data set already presented, together with a reworking of the Discussion. Reviewer 1 points out that the yeast V_1_V_0_ structure does not, apparently, show the same structural changes attributed in this work to auto-inhibition, and some discussion of this possible discrepancy, or alternative mechanism, is needed. A related point was brought up by reviewer 2, (major comments 2 and 3) asking for a clearer comparison between the mechanistic implications of the current and yeast structures, in particular addressing the possible structural changes associated with the catalytic cycle, assembly and auto-inhibition. A deeper and broader discussion that addresses these issues should make the paper have a wider impact.

Reviewer #1:

The manuscript by Kishikawa et al. determines the structures of both the intact (holo) and membrane-embedded portions of the V-ATPase from *Thermus thermophilus* in an attempt to understand the auto-inhibition of the membrane-embedded V_0_. Using Cryo-EM to determine the two structures, the authors have identified structural differences between the holo enzyme and isolated V_0_, specifically in relation to the a- and d-subunit locations, leading to their proposal that this conformational shift induces the auto-inhibition of V_0_ through the formation of a tighter V_0_ complex and side chain interactions that become possible with removal/loss of V_1_. The data and discussion provide an interesting structural model for the auto-inhibition of V_0_ and provide a framework for testing the model.

Major comments:

From the Discussion, the authors say that the isolated V_0_ structure "clearly shows the inhibitory mechanism for preventing H^+^ conductance." While the structural differences they present do suggest that the conformational changes in the a- and d- subunits are likely the major mechanism of auto-inhibition, it is not necessarily clearly defined, and their following sentence uses the phrase "most likely" which is a more appropriate description of their model. While the authors present a reasonable model for the structural changes seen between the holo V-ATPase and the isolated V_0_ that could lead to mechanical changes (Figure 6), clear evidence that these structural changes lead to auto-inhibition are not presented.

The differences between the *T. thermophilus* and yeast V-ATPase structures require more discussion than provided . The mechanism of auto-inhibition proposed by the authors is primarily the conformational changes between subunits a and d. However, the yeast V_0_ structure does not show changes in subunit d (Figure 6—figure supplement 2) and the yeast V_0_ structure showed no differences when subunit d was not present (Mazhab-Jafari et al., 2016), leading the authors of the yeast structure to state: "Consequently, the movement of the N-terminal domain of subunit a, and inhibition of proton translocation, cannot be due to interaction of the N-terminal domain of subunit a with subunit d." While the yeast V_1_V_0_ structure has not been solved, and the subunit composition differences between yeast and *T. thermophilus* may also be involved in structural changes, it is important to adequately address this incongruence between observed datasets and the model proposed in the manuscript as it is stated as a conserved model for V-ATPases.

Figure 4 nicely shows a membrane potential activation threshold for isolated V_0_. However, the results of this experiment are not fully addressed in the text.

The authors cite membrane potentials measured in *E. coli* (-75-140 mV), although higher potentials have also been proposed (ex: -220mV, Bot and Prodan, 2010). While it is unclear what potential is biologically relevant for *T. thermophilus* V-ATPase, the results in Figure 4 should be addressed in relation to what they mean for auto-inhibition of V_0_, as their preparations appear to be capable of proton pumping at greater than ~ 120 mV driving potential. Previous work with other (F-type) ATPases have also shown voltage gating, likely dependent upon F1 and F0 interactions (Feniouk et al., 2004). The voltage threshold measured for isolated V_0_ is similar to the threshold required to synthesize ATP in the holo enzyme (~110 mV) measured by the same group (Toei et al., 2007). It is therefore unclear from the results of Figure 4 why the experiment was performed and what the results mean relating to the auto-inhibition of V_0_.

Reviewer #2:

Kishikawa et al. report in the manuscript entitled "The mechanical inhibition of the isolated V_o_ from V-ATPase for proton conductance" the molecular basis of closure of the V/A-type ATPase from Thermus thermophilus. The authors solve the V_1_V_o_ complex ("holo V/A-ATPase") and the autoinhibited V_o_ domain (in absence of V_1_). They observe a different conformation of the N-terminal domain of subunit a ("a_sol_") in either states. In addition, they carry out proton conductance experiments of the isolated V_o_ domain, confirming that the isolated V_o_ domain is not able to translocate protons under physiological conditions. I think that the data described in this manuscript are important and should be considered for publication in *eLife* after substantial revisions. The authors should add additional and revise parts of their analysis (comparison of the V_o_ domains in the holo and isolated state) and take into account previously published literature on eukaryotic V-type ATPases, compare these data with the results obtained and draw general conclusions by these comparisons.

General comments:

English should be slightly improved throughout the whole manuscript. The paper is not well referenced, as the authors omit references in many places again to link directly to the relevant literature, which makes an evaluation complicated, as again the corresponding reference must be looked up. Furthermore, the authors neglect relevant literature. For example, they write "It is likely that the dissociated V_o_ loses the ability to translocate protons as a result of auto-inhibition." but do not refer to the relevant reference (Zhang et al., JBC, 1992; https://www.jbc.org/content/267/14/9773.short). Furthermore, previously released structural (e.g. Murata et al, 2005, Murata et al., PNAS, 2011, Abbas et al., 2020, Vasanthakumar et al., 2019, Tani et al., Microscopy, 2013) and computational (e.g. Krah et al., 2019) data on the V-ATPase are neglected, as well as early observations in evolutionary related F-type ATP synthases (e.g. Pogoryelov et al.,2010 (analogous to Murata, PNAS, 2011), Symersky et al., 2012), discussing possible mechanistic modes of action of proton translocation for these enzymes. It has also been proposed earlier that the N-terminal domain of subunit a adopts an alternative confirmation during the apo and holo state binding to V_1_ in the micro molar range (Oot and Wilkens, 2012).

Additional major comments:

1) "The side chain of d/R59 1 likely forms π-π stacking with a/R103"

I would expect a repulsion due to the positive charge of both residues, instead of stabilization by "π-π stacking". A closer look into the chemical environment near these residues may be required and re-modelling of the side chains may be necessary.

2) "Our structure of the isolated V_o_ suggests that the rotation of c12 rotor ring relative to the stator is mechanically hindered by a defined interaction between the a_sol_ and d subunit."

The authors suggest that the rotor is not able to rotate because it is "mechanically hindered by a defined interaction between the a_sol_ and d subunit" and thus ions (protons) cannot be transducted. However, it should be taken into account that the energy transduction of the catalytic event (hydrolysis) in eukaryotic V_1_ drives the rotation of the c-ring and the conformational change of the N-terminal domain of subunit a may be caused by assembly/disassembly of V_o_ and V_1_ (see e.g. Oot and Wilkens., 2012). In addition, computational data suggest that the c-ring is locked in an ion-locked conformation while the luminal channel (in yeast V-ATPase) is not accessible (Krah et al., 2019), also discussed in the reports of the cryo-EM structures (e.g. Mazhab-Jafari et al., 2016). If this also is likely for V/A-type ATPases, would be needed to be tested (e.g. calculation of the solvent accessible surface area for the periplasmic half-channel in the isolated V_o_ state; the holo V/A-type ATPase has been proposed to be accessible to water Zhou and Sazanov, 2019).

3) The authors further conclude: "Together, the observed results strongly suggest that the a_sol_ of the a subunit and the d subunit, absent in Fo and hallmarks structure of the V type ATPases, are key for mechanical inhibition of proton conductance through V_o_.". Together with the above-mentioned previously published data, I think that "a_sol_" and d may have an essential function, but likely involved in assembly of V_o_ with V_1_. For further conclusions about mechanistic features, the authors should also calculate the solvent accessible surface area for both half-channels of the final models (isolated V_o_ and V_1_V_o_ complex) to study if the channels are open (accessible to water) or closed (not accessible to water and other protein residues, proton transfer via water wire or side chains (reduced side chain flexibility in a dry environment) unlikely). In addition, the authors should also present other structural differences in V_o_ (if any) for holo (V_1_V_o_) and isolated V_o_ state, as e.g. differences at the a/c interface (in addition to the data shown in the SI). If there are differences found in comparison to previously published data on eukaryotic V-type ATPases, these discrepancies should be discussed, taking into account that V/A-type ATPases are also able to synthesize ATP (previous work in the same lab).

4) The Discussion section should be also adapted based on the above said.

[Editors' note: further revisions were suggested prior to acceptance, as described below.]

Thank you for resubmitting your work entitled "Mechanical inhibition of isolated V_o_ from V/A-ATPase for proton conductance" for further consideration by *eLife*. Your revised article has been evaluated by Olga Boudker (Senior Editor) and a Reviewing Editor.

The manuscript has been improved but there are some remaining issues that need to be addressed before acceptance, as outlined below:

Please respond in the Discussion to the comment from reviewer #2, specifically, "…the chemical environment around these residues should be described in detail. Alternatively, this sentence could be removed from the manuscript." The text does not have to fully agree with the reviewer, but should directly address the issue. Also, as noted by reviewer 1, some general editing for spelling/grammar.

Reviewer #1:

The revised manuscript "Mechanical inhibition of isolated V_o_ from V/A-ATPase for proton conductance" by Kishikawa et al. provides structural data for both the holo V_1_V_o_ and inhibited V_o_ ATPase from *T. thermophilus* and uses structural comparison to attempt to elucidate mechanism(s) to explain autoinhibition of the V_o_ complex. The authors also use their structural data to compare with other available structures of V ATPases to discuss the structural similarities and differences across organisms.

The authors have presented their findings in a thoughtful, structured manner with a focus on how their findings enhance the greater understanding of this protein complex. While the discussion touches upon the evolutionary differences and significance, they authors have done an admirable job to not over-interpret the data that they have collected, but rather suggest further mechanistic studies that can help support or disprove hypotheses based on their structures. The authors have also improved the manuscript in revision by minimizing repetition of findings, which improves the structure of the manuscript.

Reviewer #2:

The manuscript by Kishikawa et al. "The mechanical inhibition of the isolated V_o_ from V/A-ATPase for proton conductance" has been significantly improved and most of my concerns have been addressed. The work is important and should be published in *eLife*. However, there are still two issues which should be discussed/corrected :

1) The authors wrote: "As the reviewer pointed out, the two arginines are generally repulsive to each other. However, as shown in the added references, in a number of cases arginine is known to be involved in the enzymatic function by interacting with each other. Furthermore, our model is supported by comparison with the model reported by Sazanov et al. (PDBID: 6QUM)."

The arginine – arginine interaction is still not sufficiently described. Reference 33 is claiming that in ~90 % of the structures, which show interactions of positively charged residues are stabilized by counter charges (such as carboxylate groups).

I also had a look at the structure of the holo-V/A ATPase (pdb-ID: 6QUM). In this structure two arginine residues (aR563 and aR622) are very near, but stabilized by counter charges of aE627 and cE63. However, in case of residues aR59 and dR103 this close distance cannot be observed in the Sazanov structure (pdb ID: 6QUM). The distance of the guanidinium groups of both residues is larger than 10 Å (no interaction). In addition, in the vicinity of aR59 and aR103 several carboxylate side chains can be found, which may interact with both guanidinium groups. In addition, aR59 and aR103 are at the surface of the protein (pdb-ID: 6QUM) and thus I would expect them to be solvated if not bound to carboxylate groups.

Thus, I still think that the chemical environment around these residues should be described in detail. Alternatively, this sentence could be removed from the manuscript.

---

## [Author Response]

Reviewer #1:[…] Major comments:From the Discussion, the authors say that the isolated V_0_ structure "clearly shows the inhibitory mechanism for preventing H^+^ conductance." While the structural differences they present do suggest that the conformational changes in the a- and d- subunits are likely the major mechanism of auto-inhibition, it is not necessarily clearly defined, and their following sentence uses the phrase "most likely" which is a more appropriate description of their model. While the authors present a reasonable model for the structural changes seen between the holo V-ATPase and the isolated V_0_ that could lead to mechanical changes (Figure 6), clear evidence that these structural changes lead to auto-inhibition are not presented.

We agree with the reviewer's comments. Our structures and data do not direct evidence that the interaction between *a* subunit and *d* subunit is a major factor in the inhibited mechanism in the isolated *Tth* V_o_. Therefore, we rewrote the discussion entirety focusing on structure changes of *a*_sol_ and *d* subunit induced by association of V_1_ in the discussion part. Also, we mention not only the interactions between the *a* and *d* subunits, but also the interaction between *a* and *c*-ring in the membrane embedded region. We also indicated some points that need to be clarified in the future.

Furthermore, we carefully discussed the suggestion that the conformational changes in *a* subunit and *d* subunit were common across species and raised the possibility that these conformational change might be a conserved auto-inhibition mechanism.

“The structure of the isolated *Tth* V_o_ obtained clearly shows how it adopts a structure different from the V_o_ moiety in the holo-complex. From structural comparison between isolated V_o_ and the *holo* complex, it can be suggested that structural changes in isolated V_o_ observed in two subunits were most likely induced by dissociation of the V_1_ domain from V_o_.”

“On the other hand, Qi et al. reported that yeast V_o_ was impermeable to proton even when the interactions between the *a*-subunit and *d*-subunit were absent. These findings suggest that the interactions between *a*_sol_ and *d*-subunit are not the only mechanism by which proton permeability is inhibited. In fact, salt bridges between the arginine residues (*a*/R563, R622 in *Tth* V_o_, *a*/R735, 799 in yeast V_o_) and the glutamate residue (*c*/E63 in *Tth* V_o_, *c*/E108 in yeast V_o_) are identified in both isolated V_o_ from *T. thermophilus* and yeast V_o_, respectively. These salt bridges between the stator *a* subunit and the rotor *c*-ring inhibit proton permeability by hindering *c*-ring rotation. It is still controversial whether the formation of this salt bridge represents a *bona fide* process of proton translocation that links deprotonation and re-protonation of glutamate residues in the *c* subunits. Undoubtedly, the salt bridge must be broken by the *c*-ring rotation driven by *pmf* across the membrane or ATP hydrolysis in V_1_ to perform the functions of ATP synthesis or proton pumping (Figure 5).”

“Further studies, such as computational MD simulation, are required to assess the extent of contribution of each interaction to the auto-inhibition mechanism of *Tth* V_o_.”

The differences between the T. thermophilus and yeast V-ATPase structures require more discussion than provided. The mechanism of auto-inhibition proposed by the authors is primarily the conformational changes between subunits a and d. However, the yeast V_0_ structure does not show changes in subunit d (Figure 6—figure supplement 2) and the yeast V_0_ structure showed no differences when subunit d was not present (Mazhab-Jafari et al., 2016), leading the authors of the yeast structure to state:

We discussed the differences between V_o_ from *T. thermophilus* and eukaryotes in more detail and mentioned V_1_ induced structural change of *d* subunit in Discussion section. Please read the changes below and the responses to the first major comment.

“The *d* subunit from the mammal *holo* V-ATPase adopts a more open conformation than the yeast *d* subunit from the isolated V_o_ complex, as seen in the *holoTth* V/A-ATPase. In addition, Abbas et al. suggest that the *d* subunit from yeast *holo* V-ATPase is also more open. These results indicate that the *d* subunit in eukaryotic V-ATPase also shows the conformational change between isolated V_o_ and *holo* enzyme.”

“As described above, eukaryotic and prokaryotic V/A-ATPases appear to share a similar mechanism of conformational change at the V_o_ moiety that is advantageous for preventing proton leakage from cells or acidic vesicles. Nevertheless, there exist some interactions unique to *Tth* V_o_, as described in this paper (Figure 4—figure supplement 1), and to yeast V_o_, as reported by previous papers.”

"Consequently, the movement of the N-terminal domain of subunit a, and inhibition of proton translocation, cannot be due to interaction of the N-terminal domain of subunit a with subunit d." While the yeast V_1_V_0_ structure has not been solved, and the subunit composition differences between yeast and T. thermophilus may also be involved in structural changes, it is important to adequately address this incongruence between observed datasets and the model proposed in the manuscript as it is stated as a conserved model for V-ATPases.

As mentioned in our response to the first comment, we carefully discussed the similarities and the differences between *T. thermophilus* V_o_ and yeast V_o_. Also see our first response

Figure 4 nicely shows a membrane potential activation threshold for isolated V_0_. However, the results of this experiment are not fully addressed in the text.The authors cite membrane potentials measured in *E. coli* (-75-140 mV), although higher potentials have also been proposed (ex: -220mV, Bot and Prodan, 2010). While it is unclear what potential is biologically relevant for T. thermophilus V-ATPase, the results in Figure 4 should be addressed in relation to what they mean for auto-inhibition of V_0_, as their preparations appear to be capable of proton pumping at greater than ~ 120 mV driving potential. Previous work with other (F-type) ATPases have also shown voltage gating, likely dependent upon F1 and F0 interactions (Feniouk et al., 2004). The voltage threshold measured for isolated V_0_ is similar to the threshold required to synthesize ATP in the holo enzyme (~110 mV) measured by the same group (Toei et al., 2007). It is therefore unclear from the results of Figure 4 why the experiment was performed and what the results mean relating to the auto-inhibition of V_0_.

The structure of V_o_ determined in this study was similar to that of yeast V_o_, suggesting that Tth V_o_ also adopted similar to the yeast Vo inhibited form for proton conductance. However, the inhibition of proton conductance in isolated Tth V_o_ had not been reported yet. Considering that the membrane potential of bacteria varies depending on the environment, we focused on the pmf required for ATP synthesis (-110 mV), which value is close to the voltage threshold for proton conductance (-120 mV), and assumed that the voltage threshold is might be important to maintain sufficient pmf for ATP synthesis. The discussion has been expanded in the Results section of the manuscript. The changes are shown in red, so please refer to them.

“Previous studies have shown that isolated yeast V_o_ shows impermeability to proton, but it was unclear whether proton conductance is also inhibited in the isolated *Tth* V_o_ domain.”

“The reported membrane potential in bacterial cells varies from -75 to -220 mV depending on growth environment and methods of quantification. Although the membrane potential of *T. thermophilus* under physiological conditions is unknown, we reported previously that the *Tth* V/A-ATPase is capable of ATP synthesis when the membrane potential exceeds -110 mV. Thus, proton impermeability of the isolated *Tth* V_o_ observed at the potential less than -120 mV may function to maintain *pmf* for ATP synthesis, when *Tth* V_o_ exists solely on the cell membrane. In contrast to the V_o_ domain, several experiments have indicated that proton conductance through the bacterial F_o_ domain does not show any threshold for the membrane potential, whereas bacterial F_o_F_1_ displays the threshold likely depending on the interaction between F_o_ and F_1_. In addition, proton conductance through the F_o_ domain increases linearly with increasing *Δψ* loaded on the F_o_ liposome. These results indicate that there is no or few interactions between the *a* subunit and *c*-ring to hinder *c*-ring rotation in F_o_. Together, the observed results suggest that *a*_sol_ of the *a* subunit and the *d* subunit, that are absent from F_o_ and validated structures of the V type ATPases, can be one of the keys for mechanical inhibition of proton conductance through V_o”_

Reviewer #2:[…] General comments:English should be slightly improved throughout the whole manuscript. The paper is not well referenced, as the authors omit references in many places again to link directly to the relevant literature, which makes an evaluation complicated, as again the corresponding reference must be looked up. Furthermore, the authors neglect relevant literature. For example, they write "It is likely that the dissociated V_o_ loses the ability to translocate protons as a result of auto-inhibition." but do not refer to the relevant reference (Zhang et al., JBC, 1992; https://www.jbc.org/content/267/14/9773.short). Furthermore, previously released structural (e.g. Murata et al, 2005, Murata et al., PNAS, 2011, Abbas et al., 2020, Vasanthakumar et al., 2019, Tani et al., Microscopy, 2013) and computational (e.g. Krah et al., 2019) data on the V-ATPase are neglected, as well as early observations in evolutionary related F-type ATP synthases (e.g. Pogoryelov et al., 2010 (analogous to Murata, PNAS, 2011), Symersky et al., 2012), discussing possible mechanistic modes of action of proton translocation for these enzymes. It has also been proposed earlier that the N-terminal domain of subunit a adopts an alternative confirmation during the apo and holo state binding to V_1_ in the micro molar range (Oot and Wilkens., 2012).

As suggested by the reviewer, we have reviewed the suggested references, then reconsidered the conclusions of our previous manuscript. Especially, we focused on the stator-rotor interaction in the membrane embedded region in relation to the auto-inhibition mechanism of *Tth* V_o_. The changes are described in detail below. The English of the manuscript was proofread by a scientific proofreader throughout.

“Structural analysis of several subunits and subcomplexes of V/A-ATPases has been successfully carried out. Recent advances of single particle cryogenic microscopy (cryoEM) have facilitated structural analysis of the entire *holo* complexes of prokaryotic and eukaryotic V-ATPases in several rotational states. While the structure of the isolated yeast V_o_ has been reported in several studies, a high resolution structure of the isolated *Tth* V_o_ is still unavailable, limiting understanding the mechanism of enzyme inhibition.”

“Similarly to the two-channel model described for other rotary ATPases, the two arginine residues on the MH7 and MH8 play an important role in protonation and deprotonation of the carboxy groups on the *c*_12_ ring, with the resulting rotation of *dc*_12_ driven by proton translocation from the periplasmic to cytoplasmic side.”

“In addition, a resently reported structure of the mammal V-ATPase clearly shows that *a*_sol_ is at a distance where it cannot interact with the *d* subunit. This structure suggests that a similar conformational change in V_o_ is induced by binding of the V_1_ domain in the yeast V-ATPase, as described Oot and Wilkins previously.”

“On the other hand, Qi et al. reported that yeast V_o_ was impermeable to proton even when the interactions between the *a*-subunit and *d*-subunit were absent. These findings suggest that the interactions between *a*_sol_ and *d*-subunit are not the only mechanism by which proton permeability is inhibited. In fact, salt bridges between the arginine residues (*a*/R563, R622 in *Tth* V_o_, *a*/R735, 799 in yeast V_o_) and the glutamate residue (*c*/E63 in *Tth* V_o_, *c*/E108 in yeast V_o_) are identified in both isolated V_o_ from *T. thermophilus* and yeast V_o_, respectively. These salt bridges between the stator *a* subunit and the rotor *c*-ring inhibit proton permeability by hindering *c*-ring rotation. It is still controversial whether the formation of this salt bridge represents a *bona fide* process of proton translocation that links deprotonation and re-protonation of glutamate residues in the *c* subunits. Undoubtedly, the salt bridge must be broken by the *c*-ring rotation driven by *pmf* across the membrane or ATP hydrolysis in V_1_ to perform the functions of ATP synthesis or proton pumping (Figure 5)”

Additional major comments:1) "The side chain of d/R59 1 likely forms π-π stacking with a/R103"I would expect a repulsion due to the positive charge of both residues, instead of stabilization by "π-π stacking". A closer look into the chemical environment near these residues may be required and re-modelling of the side chains may be necessary.

As the reviewer pointed out, the two arginines are generally repulsive to each other. However, as shown in the added references, in a number of cases arginine is known to be involved in the enzymatic function by interacting with each other. Furthermore, our model is supported by comparison with the model reported by Sazanov et al. (PDBID: 6QUM).

Added reference: Neves, M. A., Yeager, M. and Abagyan, R. Unusual arginine formations in protein function and assembly: rings, strings, and stacks. J. Phys. Chem. B 116, 7006-13 (2012)

2) "Our structure of the isolated V_o_ suggests that the rotation of c12 rotor ring relative to the stator is mechanically hindered by a defined interaction between the a_sol_ and d subunit."The authors suggest that the rotor is not able to rotate because it is "mechanically hindered by a defined interaction between the a_sol_ and d subunit" and thus ions (protons) cannot be transducted. However, it should be taken into account that the energy transduction of the catalytic event (hydrolysis) in eukaryotic V_1_ drives the rotation of the c-ring and the conformational change of the N-terminal domain of subunit a may be caused by assembly/disassembly of V_o_ and V_1_ (see e.g. Oot and Wilkens., 2012).

As the reviewer suggested, we proposed the interactions between *a*_sol_ and *d* subunit are disrupted by the conformational change induced by binding V_1_ in both eukaryotic and prokaryotic V-ATPases. This idea is summarized in Figure 6 and the Discussion section. In addition, *c*-ring rotation driven by ATP hydrolysis in V_1_ might disrupt the interactions between the *a* subunit and the *c*-ring. This is also mentioned in the Discussion section. In the revised manuscript, we carefully discussed the common features and differences between yeast and *T. thermophilus* V_o_. Therefore, the Discussion section has been re-written in line with the reviewer’s suggestions. Please read our response to the General comments and the changes below.

“In addition, a resently reported structure of the mammal V-ATPase clearly shows that *a*_sol_ is at a distance where it cannot interact with the *d* subunit. This structure suggests that a similar conformational change in V_o_ is induced by binding of the V_1_ domain in the yeast V-ATPase, as described Oot and Wilkins previously.”

“The *d* subunit from the mammal *holo* V-ATPase adopts a more open conformation than the yeast *d* subunit from the isolated V_o_ complex, as seen in the *holoTth* V/A-ATPase. In addition, Abbas et al. suggest that the *d* subunit from yeast *holo* V-ATPase is also more open. These results indicate that the *d* subunit in eukaryotic V-ATPase also shows the conformational change between isolated V_o_ and *holo* enzyme.”

In addition, computational data suggest that the c-ring is locked in an ion-locked conformation while the luminal channel (in yeast V-ATPase) is not accessible (Krah et al., 2019), also discussed in the reports of the cryo-EM structures (e.g. Mazhab-Jafari et al., 2016). If this also is likely for V/A-type ATPases, would be needed to be tested (e.g. calculation of the solvent accessible surface area for the periplasmic half-channel in the isolated V_o_ state; the holo V/A-type ATPase has been proposed to be accessible to water Zhou and Sazanov, 2019).

The reviewer is right, it has been suggested that in the yeast V_o_, the luminal half channel is closed and that the channel opens transiently during the reaction. In contrast, in the prokaryotic V/A-ATPase, it has been suggested that both sides of the half channel are open. The membrane domain of the *a* subunit, which mainly composes the half channels, is very similar in the isolated *Tth* V_o_ to that of *holo* enzyme (r. m. s. d = 0.82 Å). Therefore, the half channels are mostly open in the isolated *Tth* V_o_. These differences might affect the auto-inhibition of proton translocation. Based on the comparison, we modified the manuscript and stated the issues in the Discussion section as below.

“In the isolated yeast V_o_, the luminal half-channel for releasing translocated protons to the lumen of acidic vesicles is closed and it is assumed to open transiently during catalysis. In the case of *Tth* V/A-ATPase, both sides of the half-channel are not enclosed. The membrane domain of the *a* subunit from the isolated *Tth* V_o_ is mostly identical to that of the *holo* enzyme (r. m. s. d. = 0.82 Å for A327-E637 of *a* subunit), thus the half-channels are likely open in *Tth* V_o_ as observed in *holo* enzyme. This indicates that *Tth*V_o_ is more proton permeable than yeast V_o_. This difference might be derived from the difference between ATP synthesis and ATP driven proton pump, as suggested. Further studies, such as computational MD simulation, are required to assess the extent of contribution of each interaction to the auto-inhibition mechanism of *Tth* V_o_.”

3) The authors further conclude: "Together, the observed results strongly suggest that the a_sol_ of the a subunit and the d subunit, absent in Fo and hallmarks structure of the V type ATPases, are key for mechanical inhibition of proton conductance through V_o_.". Together with the above-mentioned previously published data, I think that "a_sol_" and d may have an essential function, but likely involved in assembly of V_o_ with V_1_.

As suggested by the reviewer, this sentence ("Together, the observed results……… through V_o_."). has been excluded in the revision. In addition, we stated contribution of the salt bridges between the stator *a* subunit and *c*-ring into the inhibition of proton conductance. Please refer to our response to the General comments. As mentioned in the last paragraph of our Discussion section, we recognize the importance of *a*_sol_ and *d* subunit for assembly with V_1_. however, here we focus on the conformational changes of both *a*_sol_ and *d* subunit induced by binding of V_1_ to V_o_ rather than on their role in the complex assembly.

For further conclusions about mechanistic features, the authors should also calculate the solvent accessible surface area for both half-channels of the final models (isolated V_o_ and V_1_V_o_ complex) to study if the channels are open (accessible to water) or closed (not accessible to water and other protein residues, proton transfer via water wire or side chains (reduced side chain flexibility in a dry environment) unlikely). In addition, the authors should also present other structural differences in V_o_ (if any) for holo (V_1_V_o_) and isolated Vo state, as e.g. differences at the a/c interface (in addition to the data shown in the SI). If there are differences found in comparison to previously published data on eukaryotic V-type ATPases, these discrepancies should be discussed, taking into account that V/A-type ATPases are also able to synthesize ATP (previous work in the same lab).

As it is indicated by the reviewer, the luminal half-channel is enclosed in the isolated yeast V_o_ (Mazhab-Jafari et al., 2016, Krah et al., 2019). In contrast, both sides of the half-channel are open in *holoTth* V/A-ATPase (Zhou and Sazanov, 2019). As mentioned in the Results section (sub section “Structure comparison of the isolated V_o_ with the *holo* complex”), there are no obvious dissimilarities except for the minor differences mentioned in the manuscript. Considering the high similarity between the isolated *Tth* V_o_ and V_o_ in *holo* enzyme, we have concluded that the half-channels are also open in the isolated *Tth* V_o_, and stated this in the Discussion section of the manuscript. Please refer to our answer to the 2) comments.

In addition, we have discussed common features and differences between the prokaryotic and eukaryotic isolated V_o_ in the Discussion section. The reviewer 2 suggested structural comparisons of the isolated V_o_ and *holo* enzymes to discuss the factors that create different functional requirements for ATP synthesis or hydrolysis. However, in our manuscript, the obtained map of V_o_ domain in the *holo* enzyme was not sufficient to build an atomic model. Also, the high resolution structure of yeast V-ATPase and mammalian isolated V_o_ are lacking. Therefore, we would like to withhold the discussion of comparison with the eukaryotic *holo* enzymes.

4) The Discussion section should be also adapted based on the above said.

We rewrote the Discussion section as mentioned above.

[Editors' note: further revisions were suggested prior to acceptance, as described below.]

Reviewer #2:The manuscript by Kishikawa et al. "The mechanical inhibition of the isolated V_o_ from V/A-ATPase for proton conductance" has been significantly improved and most of my concerns have been addressed. The work is important and should be published in eLife. However, there are still two issues which should be discussed/corrected :1) The authors wrote: "As the reviewer pointed out, the two arginines are generally repulsive to each other. However, as shown in the added references, in a number of cases arginine is known to be involved in the enzymatic function by interacting with each other. Furthermore, our model is supported by comparison with the model reported by Sazanov et al. (PDBID: 6QUM)."The arginine – arginine interaction is still not sufficiently described. Reference 33 is claiming that in ~90 % of the structures, which show interactions of positively charged residues are stabilized by counter charges (such as carboxylate groups).I also had a look at the structure of the holo-V/A ATPase (pdb-ID: 6QUM). In this structure two arginine residues (aR563 and aR622) are very near, but stabilized by counter charges of aE627 and cE63. However, in case of residues aR59 and dR103 this close distance cannot be observed in the Sazanov structure (pdb ID: 6QUM). The distance of the guanidinium groups of both residues is larger than 10 Å (no interaction). In addition, in the vicinity of aR59 and aR103 several carboxylate side chains can be found, which may interact with both guanidinium groups. In addition, aR59 and aR103 are at the surface of the protein (pdb-ID: 6QUM) and thus I would expect them to be solvated if not bound to carboxylate groups.Thus, I still think that the chemical environment around these residues should be described in detail. Alternatively, this sentence could be removed from the manuscript.

We agree with the reviewer’s suggestion that the chemical environment around two arginines should be described. However, the limited resolution of our structure makes further discussion of the chemical environment difficult, and further experiments, such as mutagenesis, are needed to investigate the role of these residues. Therefore, we removed the sentence from our manuscript. And, we also modified Figure 3D along with the removal.